# SELECT: A Large-Scale Benchmark of Data Curation Strategies for Image Classification

**Benjamin Feuer**[*1]**, Jiawei Xu**[*1]**, Niv Cohen**[1]**,
Patrick Yubeaton**[1]**, Govind Mittal**[1]**, Chinmay Hegde**[1]

[1] NYU

## Abstract

Data curation is the problem of how to collect and organize samples into a dataset that supports efficient learning. Despite the centrality of the task, little work has been devoted towards a large-scale, systematic comparison of various curation methods. In this work, we take steps towards a formal evaluation of data curation strategies and introduce SELECT, the first large-scale benchmark of curation strategies for image classification.

In order to generate baseline methods for the SELECT benchmark, we create a new dataset, IMAGENET++, which constitutes the largest superset of ImageNet-1K to date. Our dataset extends ImageNet with 5 new training-data shifts, each approximately the size of ImageNet-1K itself, and each assembled using a distinct curation strategy. We evaluate our data curation baselines in two ways: (i) using each training-data shift to train identical image classification models from scratch (ii) using it to inspect a fixed pretrained self-supervised representation.

Our findings show interesting trends, particularly pertaining to recent methods for data curation such synthetic data generation and lookup based on CLIP embeddings. We show that although these strategies are highly competitive for certain tasks, the curation strategy used to assemble the original ImageNet-1K dataset remains the gold standard. We anticipate that our benchmark can illuminate the path for new methods to further reduce the gap. We release our checkpoints, code, documentation, and a link to our dataset at https://github.com/jimmyxu123/SELECT.

## 1 Introduction

Data curation is the process of collecting and organizing a corpus of data into a dataset that supports efficient learning. Until recently, data curation was an implicit consideration in most of the academic discourse on machine learning, and the vast majority of research works were oriented towards introducing novel methods, theories, or architectures.

However, data curation has begun to gain prominence as a research topic in its own right; several recent works have contended that labeling errors pervade commonly used benchmark datasets, with error rate estimates varying from 3% to 50% on the most popular ones [32, 5, 22, 28]. Group imbalances are often inadvertently introduced during the curation process, biasing model predictions [21, 10]. The work of [33] created a standard, now widely adopted, for reporting on the process for creating new datasets. Unfortunately, despite growing attention of the centrality of the data curation problem to model performance, many works in the literature do not adhere to best practices, reporting very little about the data on which they are trained, or how that data was curated [34, 24, 6]. To address

---

[*]First two authors contributed equally. Correspondence to: bf996@nyu.edu.

38th Conference on Neural Information Processing Systems (NeurIPS 2024) Track on Datasets and Benchmarks.

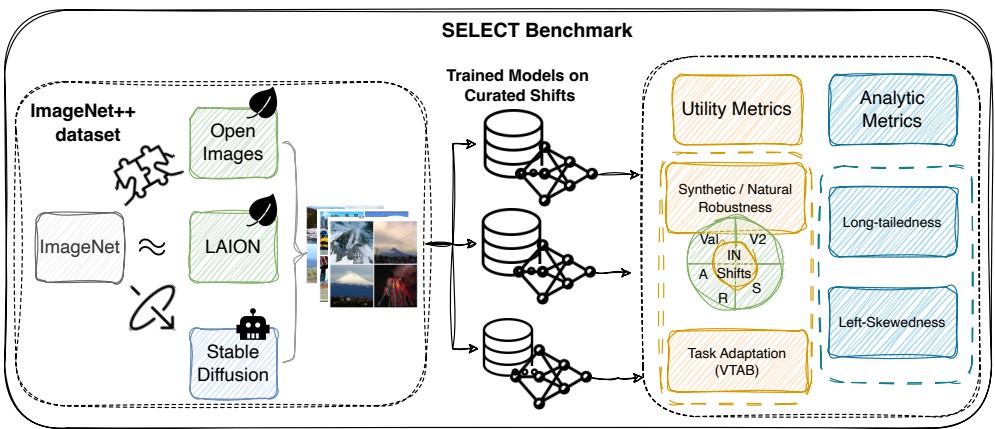

Figure 1: **Overview of SELECT Benchmark.** (**Left**) The ImageNet++ dataset is composed of different shifts of the ImageNet train set. The shifts were generated using different curation strategies and drawn from diverse data sources including OpenImages (natural images), LAION (natural images), and Stable Diffusion (synthetic images). (**Right**) We trained identical models on the sets collected using the different strategies (producing different 'shifts'), and evaluated them in two ways: (i) *Utility metrics*: quantifying the models ability to predict different in-distribution and out-of-distribution test sets, and (ii) *Analytic metrics*: examining various statistics of the distribution of the samples among the various classes.

this, [14] in NeurIPS 2023 introduced the DataComp competition, where the model architecture and loss (following CLIP [34]) were fixed and the challenge was to filter (subsample) a large pool of images to find high-performant sets of image samples for a suite of zero-shot tasks.

Our goal in this paper is to bring the implicitly studied subject of data curation into sharper focus, broaden the scope of curation beyond data filtration, and introduce it as a topic of research in its own right. In Sec. 2, we use rational choice theory to formalize any data curation strategy as a utility function, where an increment to the marginal cost produces an expected gain in utility. In Sec. 3, we introduce SELECT , a benchmark that serves as a diverse measure of utility of data curation methods in the domain of image classification. In Sec. 4, we introduce IMAGENET++ , which we leverage to produce a large-scale set of baselines for data curation, composed of 5 new training-data shifts of ImageNet-1K. Finally, in Sec. 5, we compare our IMAGENET++ baselines on the SELECT benchmark and derive several useful insights. Specifically, our contributions are as follows.

1. We introduce SELECT , a diverse benchmark for data curation methods for computer vision (in particular, image classification).

2. We introduce IMAGENET++ , the largest, most diverse set of distribution shifts for ImageNet-train to date [11]. This serves as a rich source of data curation baselines on which we train over 130 models (Fig. 2).

3. We analyze our baseline models and derive several novel insights:

   (a) On certain metrics in SELECT (pretraining and fine-tuning), reduced-cost curation methods perform as well as expert-labeled data.
   (b) However, on most metrics, expert labeling continues to outperform the alternatives.
   (c) Image-to-image curation methods generally outperform those which rely on text.
   (d) Both label noise and label imbalance remain important limiting factors on the utility of cost-efficient data-curation.

In order to enable future research and reproducibility, we release our code, our dataset, and a complete enumeration of our results for all models in the study (see supplemental attachments).

## 2 Background

### 2.1 Related Work

**Benchmarking data filtration strategies.** The work most closely related to our own is [14], a machine learning benchmark where the models are fixed and the challenge is to filter the best possible subset for pretraining. [14] compared a range of data curation strategies using standardized CLIP training code followed by a zero-shot evaluation on 38 downstream datasets. Unlike their work, ours focuses on image-only models, which are smaller and easier to train to high accuracy [13]. Our benchmark also allows for comparing a greater variety of data curation strategies and reports a more diverse range of metrics, including utility metrics on downstream tasks and analytic metrics that do not rely on model training.

**Representation learning.** The problem of data curation has long been an implicit consideration in the field of representation learning. In representation learning, images from an existing computer vision dataset are often paired with noisy labels, or replaced with some weaker form of supervision such as text captions [34, 2]. In Self-supervised learning, the labels are removed entirely and the model learns only from the images [7, 8]. Although these models require fewer labels, or in some cases no labels at all for learning a representation, they nevertheless rely on the existence of a well-defined label set for downstream tasks and benefit from a previously filtered and curated collection of images. Our work extends these efforts by illuminating the extent to which methods such as DINO are dependent on the quality of data curation at test time [7].

**ImageNet-train distribution shifts.** [40] generated a synthetic shift of ImageNet-train composed of LAION data – unlike their work, ours searches all of LAION-5B, does not rely on text similarity, and applies NSFW filtering. [40] argue that intra-class similarity of images in the original ImageNet is dramatically higher than it is for LAIONet, because searching based on an image caption alone creates an information bottleneck that mitigates the selection bias otherwise present in image-based filtering, formalizing a long-held intuition in the community that ImageNet images are stereotypical, unnatural, and overly simple representations of the class category. In a paper investigating the relationship between pretraining data diversity and fine-tuning robustness, [35] produce a 150K-sample ImageNet-like dataset using 80 diverse prompts per ImageNet class to generate the samples. Unlike their work, our dataset is publicly available, covers the entirety of ImageNet, and generates images using CLIP's image encoder only. [25] generated and released to HuggingFace 1.3 Mn images using Stable Diffusion 1.5. [1] fine-tuned a Stable Diffusion checkpoint and released the resulting dataset. [38] used Stable Diffusion 1.4 but did not release their data. Unlike previous work, our synthetic dataset was NSFW-filtered and uses CLIP's image encoder only, without any subsequent fine-tuning.

**Imbalance and quality.** The problem of label imbalance has been extensively studied in the literature, with many interventions proposed [41]. We incorporate one such intervention into our benchmark and introduce new metrics for measuring imbalance. There is also a large existing literature on detecting and correcting noisy labels; as of this writing, however, no method has been shown to work reliably across a wide range of datasets and modalities, and so we do not attempt to incorporate any label correction into our training [16]. Closely related to the concept of label fidelity is the concept of *image fidelity or diversity*. Our experiments lead us to postulate that label fidelity is necessary, but not sufficient, to achieve data diversity; as a simple counterexample, conventional data augmentation strategies such as image transformation can be designed to maintain label and image fidelity, but models trained on synthetically augmented data are only marginally more robust to natural distribution shifts [29, 42]. [36] conduct an experiment similar to our Base Accuracy experiments, but use them only to evaluate the quality of synthetic image generative models.

### 2.2 Data Curation Strategies

In this section, we formalize the problem of data curation and offer an overview of data curation strategies, as well as exemplary datasets for each curation strategy, in Tab. 1.

We model any data curation strategy as a rational series of choices made by humans with the aim of maximizing the utility of a dataset of a given size (also referred to as a shift). Through this lens, we can formalize data curation as follows.

Let $I$ be the set of plausible images (or wherever pertinent, image-text pairs). Let $D$ be a distribution over $I$. A curation strategy $f$ takes in a scalar cost input $C$, and draws a set of samples $S$ from $D$. An

increase in the cost $C$ will give a larger set $S$. Typically, for a larger $S$, the curators expect increased marginal utility on downstream tasks. For an extended discussion of cost, please refer to Sec. J.

## 3  SELECT : a benchmark of data curation for image recognition

We now introduce SELECT , the first large-scale benchmark of data curation strategies for image recognition. SELECT assumes the existence of a baseline dataset and strategy against which we wish to measure the performance of a new curation strategy. We refer to a dataset curated using a particular strategy as a *shift*. In SELECT we fix the model architecture, training strategy, and the label set, but the quantity and quality of data varies depending on the curation method. For our experiments, we fix $L$ as the 1000 labels of the ImageNet validation set, and ImageNet-train's expert curation as our baseline strategy.

We center the benchmark around ImageNet because it is arguably the most well-studied task in the vision literature for which there exist a large and growing space of OOD-robustness distribution shifts. Given how influential ImageNet has been in the development of machine learning research, we can expect the original ImageNet-1K to be a challenging baseline strategy against which to compare shifts. Nevertheless, newly curated images can outperform the labeled images in ImageNet-train, even with respect to ImageNet-val accuracy [13]. Another advantage is that ImageNet training has been heavily optimized, making the choice of a hyperparameter search space less controversial [45].

**Metrics.** The metrics in SELECT can be divided into two main categories. *Utility metrics* are designed to measure the usefulness of the curated shift for a variety of downstream tasks. Utility metrics are reliable, but more expensive to compute, since most of them require model training. *Analytic metrics* are useful for planning the data curation and rapidly evaluating different options. Therefore, such metrics are inexpensive to compute and do not require model training on the shift. They can be used to indicate the expected utility of a shift or to help explain observed differences in performance. Our analytic metrics are computed over a 1 million size sample from each shift's data, sampling uniformly and with replacement.

### 3.1  Utility metrics in SELECT

**Base Accuracy.** The first metric we report is accuracy on holdout data drawn from the same distribution as the data of the baseline strategy (in this case, ImageNet validation accuracy).

**OOD Robustness.** We report several out-of-distribution robustness metrics, both synthetic and natural. Synthetic OOD-robustness shifts are generated using algorithms which transform existing real images in the validation set (e.g., synthetic image corruptions). Natural OOD-robustness shifts contain novel real images collected according to some heuristic, such as sketches of the class [43] or only collecting class examples with unusual context [3]. For natural distribution shifts, we include *ImageNet-Sketch* [43], *ObjectNet* [3], *ImageNet-V2* - a replication of the original ImageNet test set, *ImageNet-R [19]* - a 200-class subset of ImageNet-2012, highlighting renditions of everyday objects, and *ImageNet-A* [19] a 200-class subset of ImageNet-2012 selected by misleading previous methods. For synthetic distribution shifts, we report *ImageNet-C* and *Stylized-ImageNet* [18, 15]. In Tab. 2, we report the average over all natural distribution shifts as Avg. Nat. Rob., and the average over all synthetic distribution shifts as Avg. Syn. Rob..

**Pretraining and fine-tuning.** In order to holistically assess the quality of a data curation strategy it is important to include utility metrics that do not strictly track base accuracy. One such metric is treating the model trained on the shift as a pre-trained checkpoint, and evaluating it via a diverse regime of fine-tuning tasks. Inspired by the VTAB-1k benchmark introduced in [46], we assemble 11 such tasks. The details of the tasks and our implementation details for fine-tuning models can be found in Appendix Sec. K. We report the results of such a regime in Tab. 2 as Avg. VTAB.

**Guiding self-supervised models.** All of the utility metrics we describe so far require first pretraining a model on the shift dataset. However, for rapid evaluation, it is also useful to consider metrics that estimate the utility of a curated dataset *without* training a model. For this, we turn to the field of self-supervised learning, in particular the DINO method introduced in [7]. We evaluate a DINO model pretrained on ImageNet-train; note that the DINO pretraining method makes no use of the labels in the dataset, relying entirely on the images. We then evaluate the pretrained DINO model on the ImageNet-val test set using the method of $k$NN classification described in [7]. We evaluate

the model multiple times, each time allowing a different quantity of samples-per-class (SPC). For the details of our implementation and the specific SPC values we consider, please refer to Appendix Sec. I. We report the average over all SPC values in Tab. 2 as Avg. SSL.

## 3.2 Analytic metrics in SELECT

**Summary statistics.** Summary statistics are extremely high-level features that are easy to compute and interpret. We report Dataset Size (the number of unique samples in the dataset) and class coverage (classes), which indicates the number of classes in the label set (in this case, ImageNet-1k) covered by the shift.

**Imbalance metrics.** Zipfian's distribution suggests that the frequency of an event is inversely proportional to its rank in a frequency distribution. This type of distribution is observed in natural language, where a few words (like "the" and "and") appear very frequently, while the majority are used much less often. Inspired by this law, we utilize two metrics for imbalance; one which estimates the effect of overrepresented classes, a phenomenon we term **left-skewedness**; and another estimating the impact of long-tail classes, which we refer to as **long-tailedness**. For formal definitions of these terms, please refer to Sec. G. For a visualization, please refer to Fig. 3.

**Quality metrics.** Quality metrics assess the usability of labels and images in the shift. One such score is CLIPScore, introduced in [20], uses a CLIP model (in our case, the OpenAI ViT-B-16 checkpoint) to score the similarity of image and text; this metric assesses the quality of both images and labels. Another such score is CLIP-IQA, introduced in [44], uses generic semantic opposite pairs such as good / bad and bright / dark and a CLIP model to score the quality of an image alone. We also include the extremely popular Inception Score, which measures both the diversity and recognizability of generated images by using a pre-trained Inception v3 model, with higher scores indicating better image quality and variety [37]. Finally, we include the recent CMMD score, based on richer CLIP embeddings and the maximum mean discrepancy distance with the Gaussian RBF kernel [23]. We find that Inception Score is not a reliable predictor of quality as measured by IN1000-Val accuracy, as it favors synthetic SD1000 (txt2img) images over real OI1000 and IN1000 images, which contradicts the commonsense conception of image quality as a measure of realism. CMMD score shows more promise than any other method we have considered, and has the potential to be useful; however, its low score for the OI1000 split is incongruous with other, more reliable measures of label and image quality.

**Correlational metrics.** We also provide a range of correlations which we observe to have good predictive power. All correlations are Pearson's R; precision, recall and accuracy are reported for the shift model unless otherwise specified. R:P,CC is the correlation between precision and class count. R:A,CS is the correlation between accuracy and confusion skewness (how concentrated the model error is on a few classes). R:INA,A is the correlation between accuracy of the ImageNet-1k model and the shift model. R:P,R is the correlation between precision and recall, and R: INAV, AV is the correlation between class availability in ImageNet-1k and the shift.

## 4 IMAGENET++ : A new set of baseline strategies for data curation

**Overview.** Having defined our data curation strategies in Tab. 1 and constructed a benchmark for them, we turn to producing datasets using each of our strategies and comparing them with our baseline strategy of expert curation for ImageNet-train. Few distribution shifts of ImageNet train exist, and those that exist rarely document their curation process. To fill this need, we introduce IMAGENET++ , the largest and most diverse set of shifts of ImageNet-train to date. IMAGENET++ consists of ImageNet-train and 5 distinct training shifts, each one constructed using a strategy from Tab. 1. The constituent shifts of IMAGENET++ are:

1. **OI1000:** A subset of the OpenImages dataset [26] utilizing the Crowdsourced strategy. OI1000 samples are human-labeled using crowdsourced annotators, and images are scraped without additional filtration. We assemble this dataset by creating a mapping from OpenImages to ImageNet classes and repackaging the relevant samples. The curation process required approximately 96 compute hours on CPU-only nodes.
2. **LA1000 (img2img):** A subset of the LAION dataset [39], utilizing the *Emb img2img* strategy - embedding-based search retrieving new images conditioned on each ImageNet image embedding.

Table 1: **Data curation strategies.** We enumerate the strategies we consider for curating datasets. Image quality is low for synthetic image generating methods and high otherwise, as synthetic methods can introduce noise in the image. We estimate class imbalance using our LT@500 (long-tailedness) and LS@5pct (left-skewedness) metrics, described in depth in Sec. G. We report high cost when humans were paid to label images and low otherwise. Our ImageNet label error estimates are drawn from [32], and in the absence of other information, estimates for OpenImages are assumed to be similar. Syn img2img label error estimates are identical to ImageNet, as labels are inherited.The emb-txt2img and emb-img2img label error estimates were derived from experiments in [12], who computed a 10% rate of disagreement between ImageNet original labels and those generated by a text-based embedding search (the lower bound assumes that all human errors were corrected by txt2img labeling, the upper bound assumes the union of errors). For extended definitions of the abbrevations used in this table, please refer to Sec. 2. FS:= Fully Supervised; WS:= Weakly Supervised. We enumerate the strategies employed for curating datasets.

| Strategy | Shift Name | I (image source) | T (text source) | La-bels | g (labeling function) | Filtration | Image Quality | Label Error | Imbal-ance | C (cost) |
|---|---|---|---|---|---|---|---|---|---|---|
| Expert | IN1000 | Natural | None | FS | Expert | Expert | High | 0.06 | Low | High |
| Crowdsourced | OI1000 | Natural | None | FS | Expert | None | High | 0.06 | High | High |
| Syn img2img | SD1000(**img**2img) | Model | None | WS | Algo | existing S(X,y) | Low | 0.06 | Low | Low |
| Syn txt2img | SD1000(**txt**2img) | Model | Natural | WS | Algo | Text | Low | 0.00 | Low | Low |
| Emb img2img | LA1000(**img**2img) | Natural | None | WS | Model | CLIP sim, existing S(X,y) | High | 0.04 - 0.16 | Low | Low |
| Emb txt2img | LA1000(**txt**2img) | Natural | Natural | WS | Model | CLIP sim, Text | High | 0.04 - 0.10 | Low | Low |

FS:= Fully Supervised; WS:= Weakly Supervised

The curation process required approximately 336 compute hours on a 1x-NVIDIA-RTX8000 node.

3. **LA1000 (txt2img):** A subset of the LAION dataset [39], utilizing the *Emb txt2img* strategy - embedding-based search conditioned on the CLIP similarity with the text of each ImageNet class name. This shift is an expanded version of LaionNet, originally introduced in [40].

4. **SD1000 (img2img):** A shift generated from the ImageNet-train images using a *Syn img2img* strategy; we utilize the Lambda Diffusers library from [27] to synthesize one image conditioned on each image in ImageNet-train. The curation process required approximately 672 compute hours on a 1x-NVIDIA-RTX8000 node.

5. **SD1000 (txt2img):** A shift generated from the ImageNet-train classnames using a *Syn txt2img* strategy; this is the standard process used to generate images with diffusers. The dataset was originally created by [25].

For extended descriptions of our shifts, including estimated costs of curation for each method, we refer the reader to Appendix Sec. D.

**Dataset coverage.** In this work, we generate shifts of ImageNet-train only. We do not produce new shifts for ImageNet-val, although prior works have explored this possibility, most recently [47]. We avoid this because of the high likelihood of introducing an unspecified degree of label and image noise into our validation sets. Our aim is to evaluate the training data, keeping the evaluation data as accurate as possible.

# 5   Results and Analysis

In this section, we evaluate our baseline strategy as well as our 5 shifts, training over 130 different models evaluating the utility of the shifts for different tasks. For further implementation details, please refer to Appendix Sec. L. In Tab. 2, we report our utility metrics for each strategy. Our key findings are as follows:

- **No reduced-cost curation strategy improves on ImageNet.** We explore this surprising result further in Sec. 5.1.
- **Embedding-based search strategies are the best reduced-cost curation methods.** They consistently and dramatically outperform diffusion-guided curation on most benchmarks, despite the difficulty in obtaining class-balanced data. This reinforces observations in [14] that the filtration step is particularly important when the search space is large, and in [30] that synthetic image distributions tend to saturate classifiers rapidly.

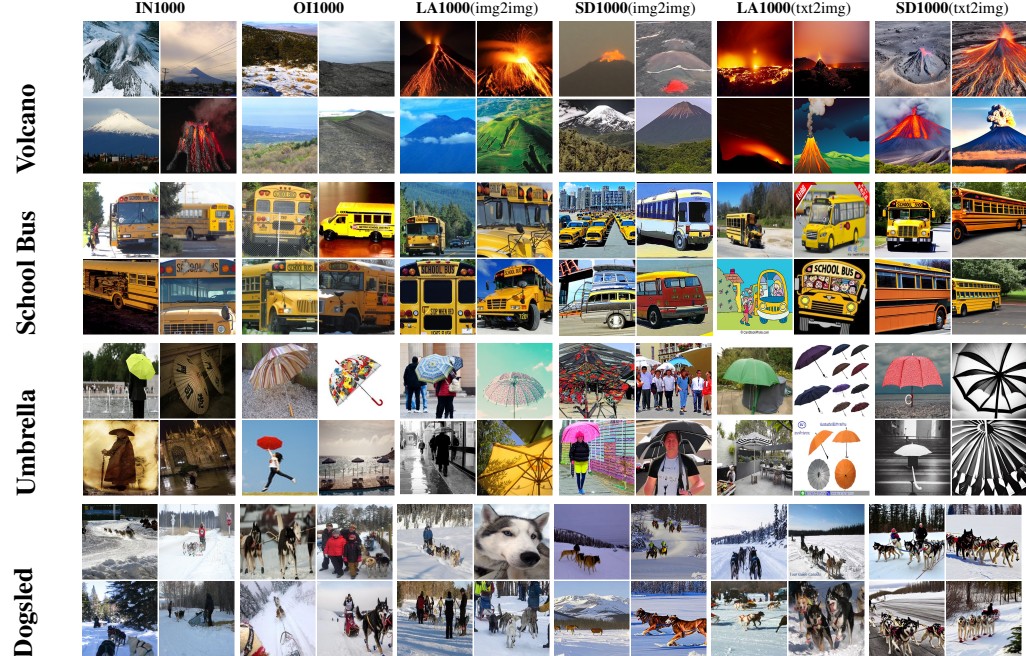

Figure 2: **Samples selected from the ImageNet++ dataset compared to those in the original ImageNet-1k dataset.** The selected classes are "Volcano", "School Bus", "Umbrella" and "Dogsled". Different viewpoints and centers emerge in these categories of LAION-1k and OI-1k. Also, samples generated in SD-1k are illustrations which may defy the laws of physics.

- **Human curation does not always lead to more useful shifts**. The OI1000 shift was expensive to curate because of its use of human annotators. However, for the majority of metrics we report, LA1000(img2img) performs better. We attribute this surprising result to label imbalance, and analyze it in greater depth in Sec. 5.2.
- **Larger datasets do not necessarily outperform smaller ones**. LA1000(img2img), despite being the smallest dataset, is the second or third best on every metric.
- **img2img strategies outperform txt2img strategies**. According to most metrics, img2img is the stronger approach; extending observations in [30] to a wider range of metrics.[2]

Table 2: **Utility of curated datasets.** Embedding search outperforms other curation methods, including the human-labeled, but heavily imbalanced, OI1000 dataset. Size is not a reliable predictor of quality; the smallest dataset is also one of the strongest performing by most metrics. Fine-tuned and self-supervised learning performance do not strictly track base task accuracy or robust accuracy, underscoring the importance of holistic evaluations of dataset quality.

| Dataset Name | Size (in M) | IN1000-Val | Avg. Nat. Rob. | Avg. Syn. Rob. | Avg. VTAB | Avg. SSL |
|---|---|---|---|---|---|---|
| IN1000 | 1.3 M | 77.9 ± 0.4 | 33.4 ± 0.5 | 23.0 ± 0.5 | 41.2 ± 0.8 | 49.8 ± 0.2 |
| LA1000 (img2img) | 0.8 M | 59.6 ± 0.4 | 23.2 ± 0.5 | 12.5 ± 0.4 | 41.1 ± 0.8 | 42.3 ± 0.2 |
| LA1000 (txt2img) | 1.0 M | 55.7 ± 0.4 | 26.0 ± 0.4 | 13.3 ± 0.3 | 39.1 ± 0.8 | 39.2 ± 0.2 |
| OI1000 | 1.2 M | 42.3 ± 0.4 | 21.8 ± 0.4 | 09.4 ± 0.3 | 44.1 ± 0.8 | 31.9 ± 0.2 |
| SD1000 (img2img) | 1.2 M | 25.2 ± 0.4 | 11.4 ± 0.4 | 04.8 ± 0.2 | 39.2 ± 0.8 | 28.8 ± 0.2 |
| SD1000 (txt2img) | 1.2 M | 23.7 ± 0.4 | 10.1 ± 0.3 | 03.3 ± 0.2 | 35.9 ± 0.8 | 35.8 ± 0.2 |

---

[2]A notable exception is distributional robustness on ImageNet shifts. We postulate that this is because the space of images is not precisely mapped by semantic descriptions of classes; terms such as "polar bear" can refer to a wide range of images with distinct features, leading to more diverse but less precise class representations. As further evidence, we note that the Avg. SSL performance of SD1000(txt2img) is strong, because even with very low SPC (samples per class), the class spaces of txt2img models are highly distinctive. However, they are not as diverse as the SD1000(img2img) class space, leading to a plateau in model performance when increasing SPC – see Appendix Sec. I.

Table 3: **Analytic metrics for curated datasets.** We consider a range of analytic metrics to help interpret the differences we observe in shift utility. The definitions for the abbreviations used in this table can be found in Sec. 3.2. For the purposes of readability, we put the quality metric headers in red. LT@500 refers to long-tailedness at 500 samples per class, and LS@5pc refers refers to left-skewedness at 5%.

| Name | Coverage | Imbalance | | Quality | | | | | | Correlation | | |
| Dataset | Classes | LT@500 | LS@5pct | CLIPScore | CLIP-IQA | Inception | CMMD | R:P,CC | R:A,CS | R:INA,A | R:P,R | R:INAV, AV |
|---|---|---|---|---|---|---|---|---|---|---|---|---|
| IN1000 | 1000 | 00.0% | 05.1% | 23.1 | 0.76 | 12.86 | 0.007 | -0.05 | -0.35 | 1.00 | 0.76 | 1.00 |
| LA1000 (img2img) | 1000 | 00.2% | 06.5% | 23.4 | 0.68 | 8.7 | 0.367 | -0.13 | -0.08 | 0.72 | 0.31 | 0.29 |
| LA1000 (txt2img) | 1000 | 58.7% | 4.5% | 23.1 | 9.97 | 0.69 | 0.251 | -0.35 | -0.11 | 0.66 | 0.32 | 0.08 |
| OI1000 | 962 | 73.4% | 74.9% | 23.5 | 0.72 | 6.79 | 0.686 | -0.24 | -0.17 | 0.37 | 0.64 | 0.06 |
| SD1000 (img2img) | 997 | 00.1% | 05.3% | 22.9 | 0.76 | 11.31 | 0.346 | 0.00 | -0.02 | 0.47 | 0.48 | 0.59 |
| SD1000 (txt2img) | 1000 | 00.1% | 05.2% | 22.8 | 0.75 | 19.09 | 0.974 | 0.00 | 0.09 | 0.36 | 0.49 | 1 |

## 5.1 Why do reduced cost strategies (still) fail to match ImageNet?

Despite innumerable advances in the field, including new vision-language foundation models, realistic generative image models and massive web-scraped datasets, it is still not possible to recreate, much less improve on, ImageNet curation without human labeling. The best human labels outperform reduced cost methods on every utility metric. In order to gain a better intuition for why reduced-cost shifts fail to match the utility of the baseline, we evaluate our analytic metrics, listed in Tab. 3.

- **img2img selection strategies exhibit strong correlation with their source dataset, harming sample diversity**. Img2img curation produces models that correlate with ImageNet in terms of per-class accuracy (See columns R:INA,A and R:INAV,AV), but fail to match it. Intuitively, this makes sense; how could a search method conditioned solely on ImageNet images produce something more diverse than ImageNet itself? Baseline datasets likely represent a performance ceiling for img2img strategies unless other factors are introduced to boost sample diversity or class balance.
- **Reduced cost methods scale noisily**. For datasets with real images, we observe that larger classes contain more label noise (Column R:P,CC). This effect is larger when the labeling strategy itself introduces noise, as is the case with embedding search based methods (LA1000) and crowdsourced labels (OI1000). Cross entropy loss has been empirically shown to be highly sensitive to label noise at high accuracy[13, 30]. Reducing label noise is an important area of improvement for embedding-based strategies. This limitation also presents opportunities – future data filtration methods could be benchmarked against their ability to denoise large classes.
- **Image and label quality metrics do not provide useful signal for guiding data curation, harming diffusion-based methods**. Diffusion models rely heavily on image quality metrics such as FID to drive progress. However, we find that the rank order agreement of these metrics with data curation utility is low, and that there is very little variance in general when comparing one shift to another. We consider the development of better metrics for these properties an important direction for future work.

## 5.2 Under-represented classes degrade utility.

Spurred by our observation that the OI1000 shift underperformed on utility metrics relative to its cost, we conduct further experiments on the adverse effects of class imbalance on model performance, and find that the presence of classes with very few samples in a dataset drives performance declines.

**Experimental details.** In this section, we use the imbalance metrics introduced in Sec. 3.2 to analyze the performance of OI1000 models when the data is blended with IN1000 data to rebalance it. To control for the possible confounding factor of label set size, we conduct this experiment at |L|=100 as well as |L|=1000.

**Results.** Tab. 4, which includes information on the data source, dataset size, our principal indicators, validation accuracy, and average accuracy under shifts, indicates that the presence of long-tailed classes is driving the performance declines in OI1000. Our indicator for long-tailedness shows strong rank-order agreement with validation accuracy. In contrast, dataset size and left-skewedness demonstrate only weak agreement.

The minimum number of samples below which a class accuracy begins to decline, however, is affected by the size of the label set and the size of the dataset. Comparing rows 8 and 9 in Tab. 4, we see that rebalancing only the classes with 101-500 samples results in an a small gain of around 6% to

Table 4: **Under-represented classes trigger performance declines.** To better understand the importance of left-skewedness and long-tailedness measures, *identical models are trained on various blended combinations of ImageNet* (green) *and OpenImages* (red) *samples*. Robustness generally correlates strongly with dataset size, unless the dataset is heavily left-skewed. Base accuracy, however, correlates most closely with long-tailedness. The classes with the longest tails ($k = 100$) are responsible for most of the performance decrease. Sample sizes are rounded to the nearest 1,000 and percentiles to the nearest whole number for clarity. The percentage in the first column indicates the proportion of data from ImageNet, while the remainder is from OpenImages. Rows are sorted by base (Val) accuracy.

| OI% | Ratio% | Dataset Size | Left-skew | Long-tail @ 500 / Long-tail @ 100 | IN100-Val / Avg. Rob. |
|---|---|---|---|---|---|
| 0% | | 125,000 | 05.0% | 00.0% / 00.0% | $85.3 \pm 0.20\%$ / $40.6 \pm 0.27\%$ |
| 38% | | 130,000 | 05.0% | 00.0% / 00.0% | $82.5 \pm 0.21\%$ / $43.1 \pm 0.27\%$ |
| 71% | | 190,000 | 45.0% | 00.0% / 00.0% | $82.2 \pm 0.17\%$ / $44.3 \pm 0.22\%$ |
| 60% | | 101,000 | 13.0% | 00.0% / 00.0% | $79.3 \pm 0.25\%$ / $41.3 \pm 0.30\%$ |
| 88% | | 90,000 | 25.0% | 00.0% / 00.0% | $76.6 \pm 0.28\%$ / $38.8 \pm 0.32\%$ |
| 67% | | 135,000 | 31.0% | 09.0% / 09.0% | $73.9 \pm 0.23\%$ / $40.7 \pm 0.26\%$ |
| 57% | | 105,000 | 12.0% | 09.0% / 09.0% | $73.4 \pm 0.27\%$ / $39.1 \pm 0.30\%$ |
| 100% | | 135,000 | 64.0% | 64.0% / 09.0% | $67.7 \pm 0.25\%$ / $37.2 \pm 0.26\%$ |
| 100% | | 53,000 | 18.0% | 67.0% / 09.0% | $58.2 \pm 0.42\%$ / $31.1 \pm 0.39\%$ |

| OI% | Ratio% | Dataset Size | Left-skew | Long-tail @ 500 / Long-tail @ 100 | IN1000-Val / Avg. Rob. |
|---|---|---|---|---|---|
| 0% | | 1,300,000 | 00.5% | 00.0% / 00.0% | $74.5 \pm 0.07\%$ / $30.7 \pm 0.08\%$ |
| 53% | | 1,200,000 | 26.0% | 00.0% / 00.0% | $69.2 \pm 0.08\%$ / $29.6 \pm 0.08\%$ |
| 28% | | 1,300,000 | 00.5% | 00.0% / 00.0% | $69.1 \pm 0.08\%$ / $31.8 \pm 0.08\%$ |
| 71% | | 1,390,000 | 34.0% | 05.0% / 00.0% | $57.7 \pm 0.08\%$ / $26.2 \pm 0.07\%$ |
| 80% | | 452,000 | 01.4% | 70.0% / 01.0% | $33.0 \pm 0.14\%$ / $13.5 \pm 0.10\%$ |
| 56% | | 683,000 | 18.0% | 75.0% / 25.0% | $30.4 \pm 0.11\%$ / $16.2 \pm 0.09\%$ |
| 100% | | 1,230,000 | 49.0% | 75.0% / 25.0% | $30.0 \pm 0.08\%$ / $17.9 \pm 0.07\%$ |
| 88% | | 1,120,000 | 43.0% | 56.0% / 06.0% | $24.4 \pm 0.08\%$ / $15.0 \pm 0.07\%$ |

accuracy, but the same change with 1000 classes (rows 13, 14) results in a much larger gain (over 25% base accuracy).

## 6   Limitations and Conclusion

In this paper, we introduced SELECT to systematically evaluate data curation strategies, curated IMAGENET++ , and evaluated 5 shifts while training over 130 models. Our analysis revealsed that cost-efficient data curation methods are growing more competitive with expert data curation methods, but that more work remains to be done to fully close the gap.

We consider this work to be an initial examination of data curation, and as such, far from complete. We report on only six curation methods, one task (image classification), and one label set (ImageNet). Another limitation of this work is that our modeling of cost is relatively coarse; as more curation methods become available and documentation of curation strategies improves, it will be possible to develop more fine-grained cost estimates. We do not ablate the choice of architecture; however, we do note that prior work has shown that this should not be expected to have an outsized effect [13]. Our analytic metrics are also constrained by the limited ability of current image quality metrics as estimators; we consider this an important area for future research.

We hope this work will spur research into new methods for data curation, and improved strategies for cost-effective data filtration, sample labeling, and synthetic data generation. In App.C we detail recommendations for future authors wishing to include data curation details in their data sheet.

## Acknowledgments and Disclosure of Funding

The authors were supported in part by the AI Research Institutes program supported by the NSF and USDA/NIFA under AI Institute for Resilient Agriculture (Award No. 2021-67021-35329), NSF grant #2154119, and by the US Department of Education's GAANN Fellowship program. Niv Cohen was partially supported by the Israeli data science scholarship for outstanding postdoctoral fellows (VATAT).

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

## A  Broader Societal Impact Statement

The goal of our work is to investigate a variety of data curation strategies and benchmark their performance on a range of downstream tasks. We do not see any negative broader societal impacts of our work that do not already exist in other methods for dataset curation, and indeed, we hope that our work will spur the advancement of more sophisticated and efficient methods for curation, which will reduce the harms incurred by employing humans to label data at scale.

## B    Licensing Statement

We release our contributions under a CC0 license (for the data we generate) and an MIT license (for our code) respectively. The remainder of the data remains bound by the rights assigned it by its original authors or license holders. We, the authors, bear responsibility in case of violation of rights and confirm that we have the right to distribute the data we have generated and the code we have written under these licenses, and that we do not have the right to modify any existing licenses binding the other code and data in our work.

## C    Reporting best practices

We recommend that authors include the following best practices as part of their data card reporting:

1. Report their approaches to the problems of data curation (See Tab. 1).
2. Benchmark some of their training data against existing curation baselines in a standardized fashion, following Tab. 2.
3. Report approximate unit costs of curating their data.
4. Provide summary metrics for their datasets, including estimates of label error and group imbalance.

## D    Extended discussion of data curation methods

### D.1    Additional details on our shifts

**OI1000.** The source of the OI1000 shift images is the OpenImages dataset introduced in [26]. Our labeling function for this shift is schema matching (with a prior of the original crowd-sourced labels). We find that 965 of 1000 ImageNet-1k classes can be matched to OpenImages equivalents via exact string matching and expert review of the label sets; the mapping we utilize is included as an artifact with this paper.
*Properties of OI1000.*    Because the images in this shift are scraped from the internet without additional filtration, this shift contains extreme class imbalances.
*Curation costs in OI1000.*    The major drivers of curation cost in OI1000 are the crowdsourced labelers.

**LA1000(img2img).**    The source of the images in this shift is the LAION-5B data from [39]. These images are unlabeled; however, descriptive text captions accompany the images. Our selection strategy for LA1000(img2img)is embedding search; using the CLIP Retrieval package, we select the samples with the greatest top-1 similarity to the images in the ImageNet-1k training set [4]. We prefilter our selections to eliminate likely duplicate images and to eliminate NSFW content.
*Properties of LA1000.*    LA1000(img2img) avoids introducing another potential confound – the text tower of the CLIP retriever. This strategy also replicates easily on datasets where labels do not translate very well to natural language representations, such as MNIST or Country211. Since labels are inferred, all embedding-based search methods introduce some label noise. *Curation cost of LA1000.*    The cost of LA1000 shifts is dominated by building and updating the large embedding tables used for the search and pretraining the large CLIP models used to search them on images and texts. These up-front costs are amortized over many datasets, so the marginal cost of LA1000 is relatively low.

**SD1000(img2img).**    Our SD1000 synthetic img2img pipeline transforms ImageNet-1k images through a one-to-one inversion process, mirroring the data from the ImageNet-1k set, thereby providing a unique perspective on image representation. We generate our synthetic images conditioned on CLIP's *image* encoder and do not use any *text* encoder; to the best of our knowledge, we are the only paper to produce distribution shifts of ImageNet-train in this manner.
*SD1000 properties.*    As all the samples in SD1000 are generated by AI, SD1000 contains image noise in the dataset.
*Curation cost of SD1000.*    The cost of SD1000 shifts is dominated by assembling the pretraining datasets for the diffusion models and training those models; these up-front costs can also be amortized over many datasets, and human labels are not required for pretraining, so the marginal cost of LA1000 is relatively low.

### D.2 Extended discussion of data curation strategies

Numerous reduced-cost data curation strategies have been proposed in the literature, sometimes with the practical aim of making curation more affordable, and sometimes with the more research-oriented objective of better understanding the effects of dataset design choices.

In Tab. 1, we enumerate these strategies and describe their distinguishing attributes.
$D(\texttt{im}, \texttt{txt})$ represents the data distribution of the baseline strategy, $D'(\texttt{im}, \texttt{txt})$ the distribution of the shift strategy. In this context, a model is any learned representation over data, such as a diffusion model or an embedding search model.

- **Expert** is the gold-standard approach for dataset curation employed by most groups, including the authors of ImageNet. Labels are selected by experts in advance, images are prefiltered by a different group of experts, and a third group applies the correct labels to the images, usually with the oversight of the research team. Considerable effort goes towards ensuring these labels are accurate.

- **Crowdsourced** labeling describes an alternate strategy; the label set is still selected by experts, but the images are not prefiltered. Instead, a wider space of potential labels is introduced and the group applying labels to images is free to select as many labels as apply to the image. This approach reduces labeling cost but can introduce class imbalance.

- **Schema matching**, used to create our OI1000 shift, describes any dataset curation strategy which seeks to find a mapping between datasets via the label set alone. The production of the schema itself is typically low-cost, but it does depend on the prior existence of well-curated origin and update datasets, and in some cases, experts to generate the mapping.

- **Synthetic** datasets generate $S(X, y)$ with AI, typically GANs or diffusion models, using as a prior some element(s) of a source dataset, such as the label set, text captions or images. We find that low $x, x_{test}$ fidelity is common when generating synthetic data via diffusion models, violating assumption (C); remedying this problem is an important direction for future work.

- **Embedding search** methods generate S via nearest neighbors search over an index of embeddings generated by a computer vision model (typically a vision-language classifier such as OpenAI's CLIP [34]. One challlenge of such methods is that they introduce label noise into the dataset. Many authors in the literature have proposed to correct noisy labels via a family of models $M(y) \rightarrow y_{correct}$ in the literature; we discuss some methods in Sec. 2.1. Another property of embedding-based search methods is that they are difficult to scale reliably. For LA1000-img2img, we retrieved over 1.3 million images, but were only able to construct a dataset of around 0.8 million images from them after NSFW filtering, deduplication, and broken links in the embedding lookup table. When constructing LA1000-txt2img from LaionNet, we encountered a similar rate of failure. [40]

## E   Predictive measures of fidelity under shift

In the bulk of this paper, we analyze distribution shifts in the aggregate, rather than individually. While this is reasonable, it is also important to better understand why certain pretraining datasets are more helpful for particular distribution shift benchmarks. This line of inquiry relates closely to that of [31].

The models we analyze here are our baseline models trained independently on each of our three shifts, as well as the model trained on ImageNet itself.

We select as our exemplary shift ImageNet-R [19], because performance on this shift varies widely and does not closely track base classifier accuracy. In Tab. 5, we report the ratio of ImageNet-R accuracy to ImageNet-Val accuracy (R-Acc-Pct). The widely referenced probit-scaled 'linear fit' hypothesis of [29] would predict that this metric should remain constant except at the extremes of ImageNet-Val; surprisingly, we see considerable variance in practice.

One important factor, clearly, is that ImageNet-R does not cover all of the classes in ImageNet, but only a 200-class subset of them. It is logical that models trained on highly imbalanced datasets, such

Table 5: **Correlation predicts shift accuracy.** We find that when accuracy on the training dataset is strongly correlated with accuracy on the shift dataset, *and* base accuracy is high, shift performance improves.

| Dataset | IN1000-Val | R-Acc-Pct | R-Cls-Acc | Pearson |
|---------|-----------|-----------|-----------|---------|
| IN1000  | 0.779     | 0.467     | 1.06      | 0.07    |
| LA1000  | 0.596     | 0.432     | 1.09      | 0.25    |
| SD1000  | 0.252     | 0.635     | 1.21      | 0.41    |
| OI1000  | 0.313     | 0.971     | 1.55      | 0.62    |

as OI1000, perform better when the class imbalance is reduced post-hoc. This phenomenon occurs at test time, because for ImageNet-R, the logits for the 800 classes which are not present in the test set are zeroed out, eliminating many of the most heavily over-predicted classes from consideration.

Still, this cannot be the entire explanation, since even after we correct for the class imbalance, models trained on OI1000 data still perform better on ImageNet-R.

Therefore, we introduce two other measures in this section. The first is the Pearson correlation coefficient (R) of ImageNet-Val accuracy and ImageNet-R accuracy. The second (R-Cls-Acc) is the relative performance of each model on the subset of ImageNet-R classes, expressed as a multiple of its performance on the entire validation set.

Surprisingly, we find that when combined, target class accuracy and correlation between validation accuracy and shift accuracy are good predictors of robust accuracy, making this a useful predictor of model performance under shift, when the relevant data is available.

## F   Label Noise as a Function of Class Hierarchy in ImageNet

As noted in our main paper, the negative correlation between precision and class size is stronger on LA1000(img2img) subset, which was created using an embedding search based method.

One research question we hope to answer is whether the negative correlation is stronger on classes with high conceptual similarity, sometimes called fine-grained classes, as it has been shown that vision-language classifiers tend to struggle with such classes [13].

Because ImageNet derives from WordNet, there exists a natural hierarchy of classes which we can exploit to better understand this question. We investigate all label in ImageNet and find that their WordNet concepts exist at varying depths in the hierarchy. We find that concepts are broadly distributed with respect to depth – the 104 deepest concepts in ImageNet exist 15 levels down the WordNet hierarchy, and the 76 shallowest concepts exist 6 levels down the hierarchy. All of the deepest classes correspond to animals, and all but six are mammals (mostly dogs). All but two of the shallowest classes share the WordNet superclass physical_entity.n.01. For our experiments on class hierarchy, we promote all classes 6 levels up the hierarchy (the deepest level at which all classes are represented). We leave ablations of this particular choice to future work – heuristically, we find that evaluating labels at this level of abstraction leads to interesting discoveries.

Specifically, we find that mammal, one of the most common superclasses in ImageNet, with 181 of the 1000 classes in ImageNet representing mammals, and one of the most fine-grained according to the WordNet hierarchy, is also one of the superclasses which suffers the greatest decline when transitioning from IN1000-trained classifiers (82% Accuracy on mammals) to LA1000-trained classifiers (55% Accuracy on mammals).

Although it is possible that label noise is introduced whenever the vision-language embedding model performs poorly on a class, we note that both our IN1000 and our LA1000 trained classifiers perform poorly on the machine.n.01 superclass (35% accuracy and 34% accuracy, respectively). If VL-embedding-model accuracy alone was predictive of label noise, we would expect the LA1000 model to perform substantially worse than the IN1000 model.

This evidence supports our hypothesis that growing datasets using embedding-based search methods tends to disproportionately introduce label noise in fine-grained classes where the classifier is less accurate.

Table 6: **Datasets with IN1000 Val. Acc and Avg. Rob.** We compare the model performance on different datasets by training a ResNet-50 model and measuring the validation accuracy on ImageNet-Val and the average robustness on the shift datasets. Specifically, models trained on shift combination datasets of ImageNet-1000 and ImageNet++ (line 5, 6, 7) are with a class-balanced weighted loss. We find that the weighted loss boosts the performance when the dataset is class-imbalanced.

| Dataset | Dataset Size | IN1000 Val. Acc | IN1000 Avg. Rob. |
|---|---|---|---|
| **IN1000** | 1.3 Mn | $77.9 \pm 0.36\%$ | $33.4 \pm 0.30\%$ |
| **OI1000** | 1.2 Mn | $31.3 \pm 0.41\%$ | $18.3 \pm 0.24\ \%$ |
| **LA1000** | 1.4 Mn | $59.6 \pm 0.43\%$ | $23.2 \pm 0.26\%$ |
| **SD1000** | 1.2 Mn | $25.2 \pm 0.38\%$ | $11.4 \pm 0.20\%$ |
| **IN1000 + LA1000** | 2.7 Mn | $74.7 \pm 0.38\%$ | $31.0 \pm 0.29\%$ |
| **IN1000 + OI1000** | 2.5 Mn | $77.1 \pm 0.37\%$ | $35.7 \pm 0.30\%$ |
| **IN1000 + SD1000** | 2.5 Mn | $69.7 \pm 0.40\%$ | $26.7 \pm 0.28\%$ |
| **IN2000** | 2.3 Mn | $69.2 \pm 0.40\%$ | $31.0 \pm 0.29\ \%$ |
| **IN5000** | 5.4 Mn | $72.7 \pm 0.39\%$ | $35.6 \pm 0.30\%$ |
| **IN21000 (ft)** | 14.2 Mn | $82.4 \pm 0.33\%$ | $44.6 \pm 0.31\%$ |

When using embedding-based search methods to grow a dataset, we find that per-class label noise is more likely to occur when *both* of the following conditions hold:

1. The embedding-generating classifier is less accurate than average on the class

2. The class is fine-grained (conceptually similar to others in the label set)

### F.1   What kinds of classes are hard to learn?

When examining the ten superclasses on which our classifiers are weakest, find strong agreement – four of the ten (clothing, consumer goods, gun, structure) are present on all training shifts. Agreement on the best performing classes is less strong. We also note that all but one of the superclasses (causal agent) are *non-living* and all but five of the best performing superclasses are *living*.

## G   Left-skewedness and long-tailedness

We propose left-skewedness @ k to be the percentage of samples in the dataset that belong to the most common k of classes, and long-tailed distribution to be the percentage of classes that contain fewer than $k$ samples ($\%< k$ Classes). Formally, we define them as follows –

**Left-skewedness @ k.** Suppose we have $\texttt{SORT}(R, f_D) \to R_s$ which sorts an array of labels $R$ in descending order per some function over a dataset $f_D$, with $D$ also an array. In this case, let $f_D$ be a frequency counter. So for $0 < k < 1$, left-skewedness @ k (LS) is given by:

$$LS(R_s, D, f, k) \subseteq \frac{|D[R_s[0 : k|R_s|]]|}{|D|} \tag{1}$$

**Long-tailedness @ k.** Now suppose that $0 < k < n$. Long-tailedness (LT) @ k is given by:

$$LT(D, R) := \frac{|R \ \texttt{s.t.} \quad \forall r \in R, r \in LT \iff |D_r| \leq k|}{|R|} \tag{2}$$

We set left-skewedness $k$ at .05. We explore two heuristic settings for long-tailedness $k$: 500 and 100.

## H   Weighted Loss Training

**Weighted loss training mitigates the effects of class imbalance.** While we have conducted detailed research on the indicators within the class-imbalanced dataset, the question of how to effectively

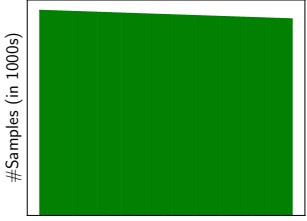
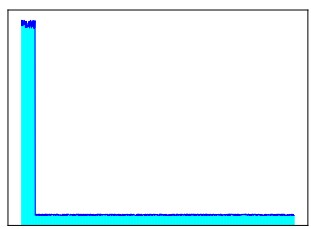
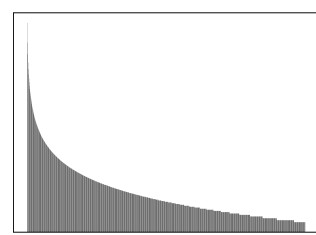

(a) **Balanced**: All classes contribute roughly equal number of samples.

(b) **Left Skewedness**: A few classes contribute most of the samples.

(c) **Long Tailedness**: Most of the classes contain few samples.

Figure 3: Illustration of Label Imbalance metrics on a dataset with 1.2 million samples and 1000 classes. **(Left)** is an approximately uniform data distribution. **(Middle)** 5% classes hold 50% samples. **(Right)** Most classes are under represented with less than 100 samples (y-axis is log-scaled).

address this issue remains open. Surprisingly, we found that a simple class-balanced weighted loss could be a powerful solution.

**Estimating the weight matrix** Given a class-imbalanced dataset $D$ of $N$ classes, we could obtain the weight matrix $W$, with each entry $W_i$ associated with the class weight of the $i$-th sample, computed as:

$$W_i = \frac{1}{f_i \cdot N} \tag{3}$$

Here, $f_i$ represents the frequency of the class to which the $i$-th sample belongs, as:

$$f_i = \frac{|D_i|}{\sum_{j=1}^{N} |D_j|} \tag{4}$$

**Computing a weighted loss** With the weighted matrix $W$ and the original loss function $l(\hat{y}_i, y_i)$ (here we choose the CrossEntropyLoss), the weighted loss for a dataset with $N$ classes can be formulated as:

$$L = \frac{1}{N} \sum_{i=1}^{N} W_i \cdot l(\hat{y}_i, y_i) \tag{5}$$

We retrain our model using a combination of ImageNet-1k and OI1000 datasets with the same parameters described in the Training setup (see Appendix Sec. L, but implemented this weighted loss function proportional to class counts. The performance improved markedly. By employing only a weighted loss, the average robustness of models across the five shift test sets increased from 18.3% to 35.7%, surpassing the baseline average robustness of 33.7% achieved when trained on ImageNet-1k (refer to Table 4). Hence, we conclude that we are able to boost the robust accuracy again with a simple intervention of reweighted loss function.

## I  Dataset curation as a guide for self-supervised pretrained models

In this section, we discuss in detail our methodology for self-supervised models, and analyze the performance of our methods on this task.

**Methodology.**  For all experiments in this section, we utilize the pretrained ResNet-50 DINO model introduced in [7]. DINO is pretrained on unlabeled samples (in this case, ImageNet-train), and at test time, is evaluated using KNN classification, guided by a fully labeled dataset.

We randomly subsample our curated datasets to acquire different numbers of samples per class (5, 10, 100 and 1000). The standard approach to KNN classification assumes that labels will be balanced; however, in real life as well as in our own curation benchmark, datasets are unbalanced and certain classes do not exist. We find that if these imbalances are treated naively, model performance suffers. Therefore, when no samples are available for a particular class, we generate random features by sampling from a Gaussian distribution in the neighborhood of a random point in latent space. If we

Table 7: **The performance of data curation methods for self-supervised guidance depends on the number of samples per class (SPC).** When only a few samples per class are provided, the highly imbalanced OI1000 is more competitive with IN1000, and SD1000(txt2img), whose prompts were highly consistent across samples, performs well. As SPC increases, the greater sample diversity of LA1000 and IN1000 allows their performance to scale better.

| Dataset | IN1000-Val (SPC=5) | IN1000-Val (SPC=10) | IN1000-Val (SPC=100) | IN1000-Val (SPC=1000) |
|---|---|---|---|---|
| IN1000 | 31.77 | 42.75 | 57.74 | 66.72 |
| LA1000 (img2img) | 27.39 | 32.10 | 50.86 | 58.91 |
| OI1000 | 24.61 | 30.19 | 31.65 | 41.10 |
| SD1000 (img2img) | 18.64 | 23.34 | 33.83 | 39.48 |
| SD1000 (txt2img) | 30.40 | 33.52 | 38.46 | 40.93 |

have some samples per class, but not enough to balance the classes, then we randomly perturb existing features by a small amount, again, sampling from a Gaussian distribution in the neighborhood of an existing real feature vector.

**Analysis.** . We observe substantial differences between our methods as a function of the number of samples per class. When only a few samples per class are provided, the highly imbalanced OI1000 is more competitive with IN1000, and SD1000(txt2img), whose prompts were highly consistent across samples, performs well. As SPC increases, the greater sample diversity of LA1000 and IN1000 allows their performance to scale better.

## J An Extended Discussion of Costs in Data Curation

**Cost.** Fundamental to understanding any utility function is the marginal cost of acquiring new samples. Unfortunately, unlike other costs incurred such as GPU training time, the cost of data acquisition is very rarely reported in the literature. In our study, we discretize our cost levels into two coarse bins: low and high. This discretization of cost precludes meaningful comparisons of marginal utility of curation functions per unit of data. Therefore, in this study we fix the quantity of data we consider and report differences in utility across curation methods. We consider the refinement of the cost variable an important direction for future research into data curation. We detail below the four main drivers of the cost in a data curation-strategy:

**Source distribution.** Also important is the choice of $I$. The typical choices are synthetic or natural; however, in real life, we choose a subset of the natural images from which to sample, and therefore condition our distribution on our data source, such as the Common Crawl [14] or Flickr [11]. If the image source is natural, there are inevitably implicit constraints, as it is impossible to effectively sample from the distribution over all conceivable images. As a general rule of thumb, we argue that more diverse and representative $D$ incur greater costs.

**Labeling function.** Once samples are collected, they must be labeled in order to use them for model training. *Data labeling* is any function $g$ which assigns labels to unlabeled samples. Traditionally, human experts have been employed for this stage. More recently, noisy labels such as image tags or captions written by humans for other purposes have been used in lieu of traditional labels. Computer vision models have also been employed to generate labels according to a variety of self-supervised strategies.

**Filtering.** After all samples have been collected, another key decision point is whether and how to subsample, or *filter*, $S$. Filtration ($F$) can take place before or after labels are produced, or at both stages. In the case of synthetic datasets, filtration can be understood as conditioning the data generating model according to some prior. Common filtration strategies include filtration via human experts (e.g., ImageNet) partial human-in-the-loop methods, and model-based methods like CLIP similarity scores. Partial human-in-the-loop includes filtering conditioned on noisy labels scraped from the web, or filtering conditioned on previously curated datasets.

**Label set.** Another important implicit decision point is when and how to introduce a *label set*, $L$. When curating a new image classification dataset, the space of available labels must be set before the labeling function $g$ is fixed, either prior to or after data filtration (unlike a zero-shot setting where $L$ is only required at test time).

# K    Impact of Shifts on Unseen Downstream Tasks

The Visual Task Adaptation Benchmark (VTAB) [46] was introduced as a way of studying how effectively pretrained models were able to adapt to unseen tasks with limited training examples. VTAB is composed of 19 downstream tasks ranging from natural images such as those found in Caltech101 to more specialized images such as those found in EuroSAT. The benchmark takes a pretrained model and finetunes it on a limited amount of training data from the downstream task before evaluating the model on the task's evaluation set. The finetuning process is fixed across datasets to ensure a fair comparison process.

We take the benchmarking process introduced in VTAB and rewrite it to be compatible with PyTorch models. In addition, we choose the following datasets for this new benchmark: Caltech256, SVHN, DTD, EuroSAT, Flowers102, Country211, FGVCAircraft, GTSRB, RenderedSST2, LFWPeople, and SUN397. We finetune each of the models introduced in this work for 1000 steps of SGD with a batch size of 64. We have an initial learning rate of 0.01 which is decreased by a factor of 10 every 300 steps of SGD. The momentum for SGD is 0.9. Results for our models and the original timm ResNet50 are present in Table 8.

We see that the vast majority of our models outperform the original timm ResNet50 model. We will first highlight two datasets where our models had similar performance to the timm model: RenderedSST2 and LFWPeople. All of the models had about a 0.5 accuracy on the RenderedSST2 dataset because it is a difficult binary classification task. None of the models presented were able to perform better than a coin-flip with the limited number of fine-tuning steps. In addition, we see that all models had a 0 accuracy on the LFWPeople dataset. This dataset is a facial recognition dataset with 5749 classes. Thus, it is understandable that the models struggled to classify the faces with such a limited number of training examples.

We see that in the SVHN and GTSRB tests, our models have about 0.7 and 0.8 accuracies respectively. However, we note that the original timm model has only 0.26 and 0.28 accuracies respectively. All of the models were given the same fine-tuning process. Therefore, the significant accuracy difference likely stems from the additional pretraining our models received from their respective data shifts. The two datasets both have relatively small label sets (10 and 43 respectively) and focus on the classification of similar image types (numbers and traffic signs respectively). Due to the way we selected images for the data shifts, it is unlikely that a significant amount of numbers/traffic sign images were added to the pretraining dataset. Therefore, our data shift likely improved the models' abilities to distinguish similar objects such as different types of traffic signs. This point is further supported by the Flowers102 and FGVCAircraft dataset (flowers and aircraft type objects respectively). We see that the accuracies for our models are significantly higher than the timm model.

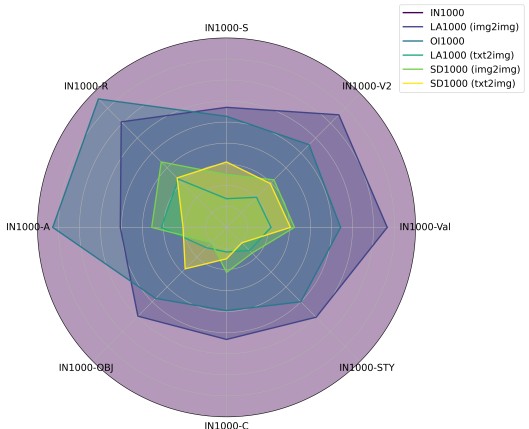

Figure 4: **The performance of various examined data curation strategies across different ImageNet evaluation sets.** Each color on the radial plot represents a different data curation strategy (Tab.1). Each direction on the plot corresponds to a distinct ImageNet evaluation set (see Sec. 3). All values are normalized based on the performance achieved with the original ImageNet training set, set as 1.0. The radial direction indicates values ranging from 0.0 to 0.9.

Table 8: **Shifts vs. Unseen Downstream Tasks.** We take the models trained on various dataset shifts introduced in this work and adapt them to 11 different datasets. The models are fine tuned for 1000 steps of SGD with a batch size of 64. The presented results are accuracy on the entire test set with 95% confidence intervals.

| Datasets | IN1000 | LA1000 (i2i) | OI1000 | SD1000 (i2i) | SD1000 (t2i) | LA1000 (t2i) | Pretrained Resnet50 (timm) |
|---|---|---|---|---|---|---|---|
| Caltech256 | $0.429 \pm 0.012$ | $0.441 \pm 0.013$ | $0.447 \pm 0.013$ | $0.319 \pm 0.012$ | $0.204 \pm 0.010$ | $0.245 \pm 0.011$ | $0.085 \pm 0.007$ |
| SVHN | $0.687 \pm 0.006$ | $0.719 \pm 0.005$ | $0.883 \pm 0.004$ | $0.698 \pm 0.006$ | $0.769 \pm 0.005$ | $0.782 \pm 0.005$ | $0.257 \pm 0.005$ |
| DTD | $0.476 \pm 0.023$ | $0.454 \pm 0.023$ | $0.397 \pm 0.022$ | $0.425 \pm 0.022$ | $0.146 \pm 0.016$ | $0.384 \pm 0.022$ | $0.273 \pm 0.020$ |
| EuroSAT | $0.957 \pm 0.005$ | $0.953 \pm 0.006$ | $0.964 \pm 0.005$ | $0.949 \pm 0.006$ | $0.814 \pm 0.010$ | $0.937 \pm 0.006$ | $0.728 \pm 0.012$ |
| Flowers102 | $0.367 \pm 0.012$ | $0.387 \pm 0.012$ | $0.371 \pm 0.012$ | $0.353 \pm 0.012$ | $0.236 \pm 0.011$ | $0.344 \pm 0.012$ | $0.113 \pm 0.008$ |
| Country211 | $0.011 \pm 0.001$ | $0.009 \pm 0.001$ | $0.016 \pm 0.002$ | $0.011 \pm 0.001$ | $0.023 \pm 0.002$ | $0.026 \pm 0.002$ | $0.007 \pm 0.001$ |
| FGVCAircraft | $0.074 \pm 0.009$ | $0.069 \pm 0.009$ | $0.103 \pm 0.010$ | $0.093 \pm 0.010$ | $0.209 \pm 0.014$ | $0.097 \pm 0.010$ | $0.017 \pm 0.004$ |
| GTSRB | $0.851 \pm 0.006$ | $0.818 \pm 0.007$ | $0.934 \pm 0.004$ | $0.811 \pm 0.007$ | $0.922 \pm 0.005$ | $0.831 \pm 0.007$ | $0.282 \pm 0.008$ |
| RenderedSST2 | $0.538 \pm 0.023$ | $0.520 \pm 0.023$ | $0.536 \pm 0.023$ | $0.530 \pm 0.023$ | $0.498 \pm 0.023$ | $0.517 \pm 0.023$ | $0.514 \pm 0.023$ |
| LFWPeople | $0.000 \pm 0.000$ | $0.000 \pm 0.000$ | $0.000 \pm 0.000$ | $0.000 \pm 0.000$ | $0.000 \pm 0.000$ | $0.000 \pm 0.000$ | $0.000 \pm 0.000$ |
| SUN397 | $0.137 \pm 0.005$ | $0.149 \pm 0.005$ | $0.200 \pm 0.005$ | $0.127 \pm 0.004$ | $0.125 \pm 0.005$ | $0.137 \pm 0.005$ | $0.026 \pm 0.002$ |

We see that our models and the timm model have similar accuracies for the DTD and EuroSAT datasets. These datasets are both "unnatural" images with DTD being composed of describable textures and EuroSAT being composed of satellite images of land. A key point in this section is that neither of these image types are found in the ImageNet-1k dataset which is used to pretrain all of the models in this table. This could suggest that the features present in these types of images are not being learned as much with our data shift. This would explain why the accuracies in our models are not significantly higher than the original timm model.

We now focus on comparing the models presented in our work. We see most of our models perform similarly with the exception of the SD1000 (t2i) model. This model tended to have significantly poorer accuracy than the other 5 models we introduce in our work. This is likely caused by the noise introduced by the Stable Diffusion generated images. This accuracy drop is likely not seen in the SD1000 (i2i) model due to the fact that we start from ImageNet-1k images instead of a text prompt. One interesting fact about SD1000 (t2i) is that it has the highest accuracy on FGVCAircraft (over 2x the next highest accuracy). The dataset is composed of various aircraft types. It is difficult to draw concrete conclusions from this result because SD1000 (t2i) performs poorly on a similar dataset (Flowers102) which is a collection of different species of flowers.

## L    Implementation Details and Extended Results for base accuracy and OOD-robustness.

**Compute.** The experiments for this work were conducted using NYU's Greene cluster. We estimate that 6422 A100-hours were used during the writing of this paper; 2190 for dataset construction, 4032 for shift model training, 200 for evaluation.

**Models trained for this paper.** We state that we train over 130 models for this paper. We arrive at that estimate in the following fashion:

- VTAB Benchmarking: 2 baselines and 5 shifts, fitted to 11 datasets -> 77 models
- SSL Benchmarking: 1 baseline and 5 shifts, fitted at 4 SPC variations -> 24 models
- Accuracy and Robustness Benchmarking: 1 baseline and 5 shifts -> 6 models
- Under-represented Classes Experiments: 17 models
- Ablation on Data Scaling (Tab. 6): 6 models

**Training setup.** We train a standard ResNet-50 with a 1000-class linear classification head in the timm library using mixed precision with a batch size of 192. The learning rate for each new combination of architecture and dataset is determined through a grid search. All models are trained for 600 epochs. Our search space is based on that of [45], with one key modification; we introduce a weighted loss function in order to compensate for class imbalances introduced by our labeling strategies. For more details on the weighted loss function, please refer to Sec. H. Our models are trained on a single node equipped with 8 AMD MI50 GPUs or 4 NVIDIA RTX8000 GPUs.

**Evaluation setup.** We evaluate our models every 10 epochs, and report all model-based results using the checkpoint with the highest accuracy on the holdout set.

Table 9: **Extended robustness results for SELECT.**

| model name | IN1000-Val | IN1000-V2 | IN1000-S | IN1000-R | IN1000-A | IN1000-OBJ | IN1000-C | IN1000-STY | Avg Nat Rob | Avg Syn Rob |
|---|---|---|---|---|---|---|---|---|---|---|
| IN1000 | 0.779 | 0.648 | 0.235 | 0.364 | 0.087 | 0.176 | 0.402 | 0.058 | 0.302 | 0.23 |
| LA1000 (img2img) | 0.596 | 0.49 | 0.134 | 0.258 | 0.044 | 0.105 | 0.214 | 0.035 | 0.206 | 0.125 |
| OI1000 | 0.423 | 0.36 | 0.124 | 0.314 | 0.072 | 0.084 | 0.159 | 0.029 | 0.191 | 0.094 |
| LA1000 (txt2img) | 0.557 | 0.469 | 0.268 | 0.405 | 0.053 | 0.105 | 0.225 | 0.041 | 0.26 | 0.133 |
| SD1000 (img2img) | 0.252 | 0.207 | 0.059 | 0.16 | 0.031 | 0.018 | 0.086 | 0.01 | 0.095 | 0.048 |
| SD1000 (txt2img) | 0.237 | 0.191 | 0.073 | 0.121 | 0.018 | 0.049 | 0.06 | 0.006 | 0.09 | 0.033 |


# M   Availability

Our entire dataset, including Croissant metadata record and our trained model checkpoints, are currently available on HuggingFace. All shifts are made available in WebDataset or HuggingFace Datasets format. The links can be accessed at our GitHub repository, https://github.com/jimmyxu123/SELECT. Our hosting and maintenance plan is to preserve the work via the HuggingFace repository, which has proven to be a reliable exchange for large datasets in recent years.

# N   Not safe for work (NSFW) filtering

The images included in ImageNet++ are sourced from the LAION-5B dataset ([39]), the OpenImages dataset ([26]), and synthetic img2img inversion transformations from the ImageNet-1k dataset. Although these datasets are generally regarded as safe and publicly available, we employ a variety of NSFW content filtering techniques to identify and exclude any potentially problematic images and captions.

Firstly, we filter captions using Detoxify ([17]), a robust language model designed to detect toxic comments. Specifically, we employ the multilingual XLM-roBERTa ([9]) variant. This model generates scores ranging from zero to one for the following categories: toxicity, severe toxicity, obscenity, identity attack, insult, threat, and sexually explicit content. Based on the prior work in image filtering by DataComp ([14]), we heuristically set a threshold of 0.1. This threshold effectively filters NSFW text while minimizing false positives. If any of the Detoxify category scores exceed this threshold, the sample is discarded. Next, we apply a filtering process to the visual data. We utilize a modified version of LAION-5B's CLIP-based binary NSFW classification model by [39], which employs CLIP ViT-L/14 visual embeddings as input. Further information about the training data is provided in Appendix C.5 of the LAION-5B paper. In summary, the dataset comprises 682,000 images, with a roughly equal distribution between Safe for Work (SFW) and NSFW categories.

After applying this filtering to the three subsets of ImageNet++, no toxic images were found, indicating that the dataset's captions are safe. However, after applying this filtering to the three subsets of ImageNet++, no toxic images were found, indicating that the dataset's captions are safe. This result isn't surprising given that the source data has been previously vetted by machine or human experts.

# O   Datasheet

**Motivation**

**For what purpose was the dataset created?**
ImageNet++ aims to facilitate the training of models robust against natural distribution shifts, efficiently utilizing data. Including three datasets, OI1000, Laion-1k, and SD1000, each introducing natural distribution shifts relative to ImageNet-1k, it is the largest and most diverse superset of ImageNet-1k. Moreover, we use ImageNet++ to derive novel insights into scaling factors in this paper.

**Who created the dataset (e.g., which team, research group) and on behalf of which entity (e.g., company, institution, organization)?**
The dataset was created by researchers in the DICE Lab at New York University.

**Has the dataset been used already? If so, where are the results so others can compare (e.g., links to published papers)?**
The dataset was used for experiments in this paper.

**What (other) tasks could the dataset be used for?**
The dataset could also be used for model pretraining. The method could also be applied to generate the same-size shifts to other datasets.

**Any other comments?** None.

**Dataset Composition**

**What do the instances that comprise the dataset represent?**
ImageNet++ consists of 5 distinct datasets, each representing a variation of the ImageNet-1k dataset:
1.OpenImages-1000(OI1000): A subset of the Open Image dataset[26], where samples are aligned with ImageNet-1k class names based on human-labeled annotations.
2.Laion-1000(LAION1000): A subset of the unlabeled LAION dataset[39], selected through nearest neighbors search against the ImageNet-1k training set.
3.Stable Diffusion-1000(SD1000): A set generated from the ImageNet-1k dataset using Stable Diffusion, where images are transformed via an inversion process.

**How many instances are there in total?**
See Table 6 for reference of our dataset.

**What data does each instance consist of? "Raw" data (e.g., unprocessed text or images)? Features/attributes? Is there a label/target associated with instances?**
Instances in OI1000 and LAION1000 are images each associated with labels and captions. SD1000 contains AI-generated features based on the images from ImageNet-1k, also with associated labels. All the included data are filtered for NSFW content (see Appendix N)

**Is any information missing from individual instances? If so, please provide a description, explaining why this information is missing (e.g., because it was unavailable). This does not include intentionally removed information but might include, e.g., redacted text.** There is no missing information in the dataset.

**Does the dataset contain all possible instances or is it a sample (not necessarily random) of instances from a larger set? If the dataset is a sample, then what is the larger set? Is the sample representative of the larger set (e.g., geographic coverage)? If so, please describe how this representativeness was validated/verified. If it is not representative of the larger set, please describe why not (e.g., to cover a more diverse range of instances, because instances were withheld or unavailable).**
Instances in OI1000 and LAION1000 are raw images, while SD1000 comprises AI-generated features derived from ImageNet-1k images. All instances are labeled. The datasets, particularly OI1000 and LAION1000, are subsets of larger sets and are intentionally curated to introduce specific feature shifts relative to ImageNet-1k, rather than to serve as comprehensive representations of their parent datasets.

**Are there any errors, sources of noise, or redundancies in the dataset? If so, please provide a description.**
There are no known errors, noise, or redundancies in the dataset.

**Any other comments?**
None.

**Collection Process**

**What mechanisms or procedures were used to collect the data? (e.g., hardware apparatuses or sensors, manual human curation, software programs, software APIs)**
All the data of OI1000 and Laion-1k are collected from larger public sets. Data in SD1000 is generated by AI.

**If the dataset is a sample from a larger set, what was the sampling strategy (e.g., deterministic, probabilistic with specific sampling probabilities)?**

1.OI1000 (OpenImages-1000): The sampling strategy was deterministic, based on a direct mapping of human-labeled class names to the corresponding classes in ImageNet-1k.

2.LAION1000: The sampling was semi-probabilistic. Samples were selected using a nearest neighbors search based on the ImageNet-1k training set. While this approach is guided by the proximity of LAION images to the ImageNet-1k feature space, it inherently introduces a probabilistic element due to the variability in nearest-neighbor results.

3.SD1000 (Stable Diffusion-1000): This subset encompasses all possible instances generated from the ImageNet-1k dataset using Stable Diffusion, hence it's not a sample but a complete set derived from the original dataset through a generative process.

**Who was involved in the data collection process (e.g., students, crowd workers, contractors), and how were they compensated (e.g., how much were crowd workers paid)?**
The creation of ImageNet++ is done by the author of this work.

**Over what timeframe was the data collected?**
The timeframe for creating the ImageNet++ is from 12/2023 to 1/2024.

**Any other comments?** None.

### Data Preprocessing

**Was any preprocessing/cleaning/labeling of the data done (e.g., discretization or bucketing, tokenization, part-of-speech tagging, SIFT feature extraction, removal of instances, processing of missing values)? If so, please provide a description. If not, you may skip the remainder of the questions in this section.**
As our images are collected either from public data sources or synthetic generation, we did an NSFW filtering on all the images and the captions (see Appendix N).

**Was the "raw" data saved in addition to the preprocessed/cleaned/labeled data (e.g., to support unanticipated future uses)? If so, please provide a link or other access point to the "raw" data.**
Yes, the "raw data" was also public.

**Is the software used to preprocess/clean/label the instances available? If so, please provide a link or other access point.**
The details can be found in Appendix N.

**Does this dataset collection/processing procedure achieve the motivation for creating the dataset stated in the first section of this datasheet? If not, what are the limitations?**
We hope that the release of this benchmark suite will achieve our goal of accelerating research in models' robustness to natural shifts, as well as making it easier for researchers and practitioners to generate data augmentations via our benchmark.

**Any other comments?** None.

### Dataset Distribution

**Will the dataset be distributed to third parties outside of the entity (e.g., company, institution, organization) on behalf of which the dataset was created?**
The dataset will be public soon. All researchers and practitioners can access it if they are interested in the dataset.

**How will the dataset be distributed (e.g., tarball on website, API, GitHub)?**
We will publish all the format of the data.

**When will the dataset be released/first distributed? What license (if any) is it distributed under?**
The dataset is public as of 6/2024.

**Are there any copyrights on the data?**
There are no copyrights on the data.

**Are there any fees or access/export restrictions?**
There are no fees or restrictions.

**Any other comments?**
None.

**Dataset Maintenance**

**Who is supporting/hosting/maintaining the dataset?**
The authors of this work are supporting/hosting/maintaining the dataset.

**Will the dataset be updated? If so, how often and by whom?** We welcome updates from the community.

**How will updates be communicated? (e.g., mailing list, GitHub)**
Updates will be communicated by the mailing list of the authors.

**If the dataset becomes obsolete how will this be communicated?**
If the dataset becomes obsolete, it can be communicated by the mailing list of the authors.

**If others want to extend/augment/build on this dataset, is there a mechanism for them to do so? If so, is there a process for tracking/assessing the quality of those contributions? What is the process for communicating/distributing these contributions to users?**
Others can publish their extends/augmentation on the benchmark to any open-source website (eg. HuggingFace, Github, etc.)

**Any other comments?**
None.

**Legal and Ethical Considerations**

**Were any ethical review processes conducted (e.g., by an institutional review board)? If so, please provide a description of these review processes, including the outcomes, as well as a link or other access point to any supporting documentation.**
There was no ethical review process. However, we did filtering for NSFW information before publishing the dataset.

**Does the dataset contain data that might be considered confidential (e.g., data that is protected by legal privilege or by doctorpatient confidentiality, data that includes the content of individuals non-public communications)? If so, please provide a description.**
All the data are either collected from public source or generated by AI. There is no confidential data.

**Does the dataset contain data that, if viewed directly, might be offensive, insulting, threatening, or might otherwise cause anxiety? If so, please describe why.**
We did NSFW filtering to prevent this problem. As we believe, none of the data might be offensive, insulting, threatening, or otherwise cause anxiety.

**Any other comments?**
None.

