# OpenReview forum: "SELECT: A Large-Scale Benchmark of Data Curation Strategies for Image Classification"
_NeurIPS.cc/2024/Datasets_and_Benchmarks_Track — NeurIPS 2024 Track Datasets and Benchmarks Poster_

### Official Review · Reviewer_kQRC · 2024-06-28
**Review ImageNet++**

**Rating:** 6
**Confidence:** 3
**Correctness:** Correctness seems to be valid.

**Review:**

The paper is a mixed bag of content, it is too much content in some parts and then not general enough in others. It is very strong sometimes but not convincing in others. In one sentence, I would describe the paper as a very detailed analysis of a small parts of the whole data curation process which makes it difficult to generalize to the whole process. The strengths and weaknesses are provided below in detail.

The biggest strength is the detailed evaluation. It discusses many parts and insights and even more in the supplement. However, this is also a drawback since the paper is cluttered with content and it makes it difficult to understand the main key results. The authors highlight some key insights but after reading the paper I could not have repeated any major point since there were so many results. My recommendation streamline the paper and focus on two or three main conclusion, move anything not relevant to these conclusions to the appendix. I know this is difficult and it happened to me more than once but it the current form it just not good to read.

The biggest issue for me is the generalizability. The authors explain in section 2.2 many important parts for the data curation process. However, while the benchmark claims to be " a diverse benchmark for data curation method" it just focuses on 5 very specific datasets alike to imagenet. Since no individual factors are adjustable, e.g. increase only the data quality it is difficult to draw general conclusions. The authors acknowledge this even partially since for example the cost is only subdivided in low and high which is a very rough which indicates to me that the authors know that a nuanced change is not considered. However, I argue having the ability to adjust these dimensions individually in a fine grained manner allows to draw general conclusions.

**Strengths:**

The evaluation and details are very good there is so much information and content in this paper. I prefer paper which include all aspects in one paper rather than trying to split them up in different papers. The data curation process in general is a very important topic and larger scale studies e.g. based on Imagenet after rarely found. The authors raise a lot of question, have a lot of interesting observations which need more detailed anaylsis.

**Additional Feedback:**

See above

**Clarity:**

The paper is well written with respect to grammer and language. It is not well written respect to mental load. It is cluttered with content which makes is difficult to find the main results. (see above for more details)

**Documentation:**

I checked them briefly and they look good.

**Ethics:**

There are no major concerncs. The paper uses already published datasets and even highlights the filtering of NSFW content.
Could you please state if or how you checked for copyright or consent?

**Limitations:**

The limitation should be described in appendix D. However, it only describes the datasets in more detail. No header or even the word "limitation" is used which would indicate a section about the limitations of this paper. This section should also be in the main paper. The supplement is just for more details while the main information (like the main drawbacks/limitations) should be in the main paper.

**Opportunities For Improvement:**

- section 2.2 is a bit superficial what is the point here, I think you can move something to the supplement here and make it shorter leaving more room for other topics
- The caption of table 1 should be more self-explained. FS and WS should be in the caption not below the table I missed it several times before noticing. Please indicate shortly the different topics since Sec 2 is quite broad and long. Based on the text I could not reproduce when something is Low or Higg with regard to Quality, Imbalance or cost. The Label Error should be explained shortly and not only referencing the source.
- Are the sample of figure 2 randomly picked or manually picked?
- Why is the supplement split between main paper and actual supplement?
- Typo on page 14, missing citation
- Please provide the full robustness scores in the appendix. Since the robustness dataset are quite diverse it would be interesting to see what different results are achieved. Moreover, the standard deviation (?) in table 2 seems to be very low for different datasets. Can you explain this?
- Can you please explain how you trained 130 models? I don't see any specifics how you reach this number?

**Relation To Prior Work:**

The related work is discussed however in my opinion the topic of other data-centric benchmarks aside from data-comp is missing. I'm aware of multiple diffferent directions which are often smaller and more narrow in scope and thus different but it would have been good to add some more here.

**Summary And Contributions:**

The authors present a benchmark about dataset curation for ImageNet. They proide 5 variants of ImageNet train on which they train multiple models and then perform a detailed analysis of these models. The contributions are the 5 variants as well as the evaluation. The authors also claim to be a benchmark for data curation in general (in particular for image classification) and novel insights based on the 5 variants.

---

> ### Author Rebuttal · Authors · 2024-08-15
>
> Thank you for your review. We are happy that you found our evaluation detailed and our insights extensive. We address each of your questions below:
>
> **W1 (Sec 2.2):** We agree with you that Section 2.2 in the main paper can be condensed. The most essential elements here are the definitions of terms and the discussion of cost, as noted by reviewer 5d11. In the camera-ready version, we will relocate the remainder of the content to the appendix.
>
> **W2 (Tab 1 Caption):** We thank you for this note about the caption of Table 1; we agree that it should be made clearer. We have altered it as follows –
>
> ```
> We enumerate the strategies we consider for curating datasets. Image quality is low for synthetic image generating methods and high otherwise, as synthetic methods can introduce noise in the image. We estimate class imbalance using our LT@500 (long-tailedness) and LS@5pct (left-skewedness) metrics, described in depth in App. Sec. G. We report high cost when humans were paid to label images and low otherwise. Our ImageNet label error estimates are drawn from [1], and in the absence of other information, estimates for OpenImages are assumed to be similar. Syn img2img label error estimates are identical to ImageNet, as labels are inherited.The emb-txt2img and emb-img2img label error estimates were derived from experiments in [2], who computed a 10% rate of disagreement between ImageNet original labels and those generated by a text-based embedding search (the lower bound assumes that all human errors were corrected by txt2img labeling, the upper bound assumes the union of errors). For extended definitions of the abbrevations used in this table, please refer to Sec. 2. FS:= Fully Supervised; WS:= Weakly Supervised.
> ```
>
> Please let us know if this will address your concerns, or if you would like additional changes.
>
> We also note that the label error field of Emb txt2img should read 0.04 - 0.1; we will correct this in the final version of the paper.
>
> **Q3 (Samples / classes of Fig 2):** Thanks for the question – the classes and samples were chosen randomly for Figure 2.
>
> **Q4 (Supplement split):** We divided our supplements because this year, the NeurIPS Datasets and Benchmarks track had separate deadlines for different types of appendix content. In the camera-ready version, they will be unified.
>
> **W5 (Missing internal link):** Thanks for pointing out the typo; we will correct it in the camera-ready draft.
>
> **W6 (Full robustness results):** We have added our full robustness scores to our global response PDF, section (A). We will also add them to the appendix of the final version of our paper.
>
> **W7 (small confidence intervals):** We thank the reviewer for catching the error in our confidence intervals, which were computed on the train splits rather than the test splits. We have updated our table with corrected intervals; the corrected table can be found in our global response PDF, section (B).
>
> **W8 (130 models):** Thanks for raising this clarifying point! Contribution 2 in our introduction states that we trained over 130 models for this paper. Here is how we arrived at the estimate;
>
> VTAB Benchmarking: 2 baselines and 5 shifts, fitted to 11 datasets -> 77 models
>
> SSL Benchmarking: 1 baseline and 5 shifts, fitted at 4 SPC variations -> 24 models
>
> Accuracy and Robustness Benchmarking: 1 baseline and 5 shifts -> 6 models
>
> Under-represented Classes Experiments: 17 models
>
> Ablation on Data Scaling (Tab. 6): 6 models
>
> As we report results for 130 models, and as we train considerably more models than that (failures to converge, coding errors, unreported results), we state that we train over 130 models.
>
> We will add the above to an appendix section in the camera-ready draft.
>
> **W9 (limitations):** Thanks for this note – we agree that a limitations section is very important, and we will add one to our camera-ready draft. In it, we will emphasize that we report on only six curation methods, one task (image classification), and one label set (ImageNet). Another limitation of this work is that our modeling of cost is relatively coarse; as more curation methods become available and documentation of curation strategies improves, it will be possible to develop more fine-grained cost estimates.
>
> **W10 (related work):** Thanks for this note – in the camera-ready draft, we will combine our extended related works section (currently in the appendix) and our main related works section.
>
> **W11 (clutter):** We thank you for this note, and agree that our results section could be made more easily interpretable. To remedy this, we propose adding a roadmap paragraph at the end of Sec 1 which will link to our various results sections and briefly summarize the important conclusions we reach. We hope that this addresses your note, but if you would prefer some other solution, please feel free to let us know.
>
> **W12 (generalizability):** We thank the reviewer for the pertinent question about how well our conclusions about trends in data curation will generalize. As we note in our introduction, data curation is understudied; for this reason, we consider the SELECT benchmark our core contribution and the ImageNet++ dataset our secondary contribution, as we believe they can drive progress in this subfield, leading to more robust analyses over time. As we describe in W9, we will add a limitations section emphasizing the data on which our conclusions are based.
>
> Thank you very much, once again, for your excellent comments. We respectfully ask that if you feel more positively about our paper, to please consider updating your score. If not, please let us know what can be further improved; we are happy to continue the discussion any time until the end of the discussion period. Thank you!
>
> **REFERENCES**
>
> [1] https://arxiv.org/abs/2103.14749
>
> [2] https://arxiv.org/pdf/2210.07396

---

> > ### Comment · Reviewer_kQRC · 2024-08-19
> >
> > Thank you for your rebuttal. You addressed the majority of minor issues raised this I increased my score.
> > Especially W10-12 are in my opinion not enough addressed.
> > W10 - I meant data-centric works outside your already mentioned domain. I have the feeling this topic is not so unexplored as you make it seem.
> >
> > W11 - this might. I feel addiotnal viusal highlights and even potentially removing some minor results would be important to improve the readability.
> >
> > W12 - I see your argument but honestly it does not convince me fully. I have the feeling you are overselling your results here. Your evaluation is limited to one dataset some variants. I skeptical about the claimed insights with such a base. You claim to be the first large scale curation benchmark. Due to your one sided related work I"m also skeptical here. These are already two.out of your three main contributions.

---

> > > ### Author Response · Authors · 2024-08-20
> > >
> > > Thanks for your note!  We are happy we were able to address most of the minor issues you raised.
> > >
> > > **W10:** Although we conducted a thorough literature review on work which directly influenced ours (I.e., ImageNet-train distribution shifts, self-supervised learning algorithms in computer vision, label imbalance, label error detection, reference-free quality metrics), it is true that there exists a large data-centric ML literature beyond our domain, such as the data quality literature in NLP. If there are any particular areas you feel we should have touched on or works we should have considered, we are happy to do so.
> > >
> > > **W11:** We are happy to consider any particular ideas for highlights or results to relocate which you may have in mind.
> > >
> > > **W12:** We respect your concerns about overgeneralization. Perhaps there is some particular language you would prefer we consider in the limitations section to better scope our findings?

---

### Official Review · Reviewer_5d11 · 2024-07-22
**Review of the SELECT Benchmark proposal**

**Rating:** 7
**Confidence:** 3
**Correctness:** The claims made by this work seem app…

**Review:**

The authors have proposed a novel contribution from an ML data-centric perspective by focusing on evaluating and studying the impact of different data curation processes in ML models for the image classification task. The contribution is novel, and some approaches included in the benchmark are based on well-known published literature.

The benchmark is showcased by creating a new dataset and expanding ImageNet with different curation strategies. Studying the results of the dataset on the benchmark, the authors raised relevant conclusions showing the utility of the proposed benchmark for analyzing curation processes.

**Strengths:**

The proposed SELECT benchmark provides a novel approach for evaluating different curation methods. Terms such as the cost of the method, often neglected, are a novel contribution.

Some methods applied to evaluate them are based on already and influential published literature.

The authors propose to include the formal definition of the benchmark as a reporting best practice enabling data documentation proposals such as data cards.

The benchmarks are showcased with a novel dataset proposed by authors from imageNet. Further analysis is drawn from the benchmark results, which show its usefulness.

**Additional Feedback:**

Extending the recommendation for responsible data reporting could be an exciting outcome of this work.

**Clarity:**

The paper is well-written, and an extensive appendix complements some proposed concepts.

**Documentation:**

The authors provide a repository containing data and the code to reproduce the benchmark. However, as the authors propose a dataset, I think there is a need to provide data documentation in a standardized format, such as Data Cards, or using some machine-readable format, such as the one proposed by the track.

**Ethics:**

There is no clear ethical issue involved.

**Limitations:**

Conclusions drawn from the benchmark results using the proposed dataset may be limited by the specific curation processes of the analyzed shifts. For instance, concluding that the cost of the curation process may not correlate with higher scores in the benchmark could be true for the analyzed shifts but could not be a “general” conclusion about the curation process. I think this needs to be clearly stated in the manuscript.

**Opportunities For Improvement:**

The conclusion about the cost seems strange to me, as more expensive curation methods do not lead to better performance; the most expensive method (the original Imagenet-1K curated by experts) remains the most performant one.

The gap between the baseline and different shifts is significant in many of the reported metrics. As the utility of imageNet++ to test and showcase the benchmark is clear, I’m not sure about the dataset’s utility as a central contribution to the paper and the community.

**Relation To Prior Work:**

Prior work are correctly stated.

**Summary And Contributions:**

In this work, the authors propose a set of metrics and assumptions for the formal evaluation of data curation strategies for image classification. They suggest a formal definition of the data curation strategy and a set of utility and analytical metrics to evaluate the dataset curation strategy over a baseline dataset. The utility metrics evaluate the base accuracy of the models trained with the data, the out-of-distribution robustness applying natural and synthetic approaches, the feasibility of adapting to a new task when used as a pre-trained model using the VTAB approach, and the utility of the curated dataset in self-supervised learning approaches.

To test the benchmark’s suitability, the authors present a new dataset extending ImagetNet using 5 different curation strategies. These curation strategies range from human (crowd workers) to automatic (synthetic data) approaches and come from different source datasets (LAION, OpenImages, or Imagenet itself).

The conclusion shows how the original ImageNet outperforms practically all the metrics in the different proposed dataset shifts. It also shows how the cost of the curation process and the size of the dataset are not a direct proxy of high values in the benchmark. Finally, it shows how synthetic txt2img outperformed my img2img strategies in the benchmark.

---

> ### Author Rebuttal · Authors · 2024-08-15
>
> Thank you for your detailed review. We appreciate that you find our results useful, and that we address the question of cost, which often goes uninvestigated as a term in the related literature. We address each of your questions below:
>
> **W1 (conclusions about the cost)**: Thanks for bringing this topic up! We assume that you are referring to our bullet points in section 5 (please feel free to reply if this is not the case), which we paraphrase below:
>
> ```
> On certain metrics, in particular Avg. VTAB, reduced-cost curation methods perform as well as the baseline. However, for the majority of metrics, as well as in the aggregate, no curation strategy outperforms the baseline. Higher cost curation strategies do not always lead to more useful shifts; I.e., our OI1000 shift is high-cost, and has lower utility than some low-cost methods. We attribute the performance gap to label imbalance.
> ```
>
> We agree that we could have been clearer, so we propose revising it to –
>
> ```
> On certain metrics, in particular Avg. VTAB, reduced-cost curation methods perform as well as the baseline. However, for the majority of metrics and cases we examine, more expensive curation methods lead to better performance. One notable exception is our OI1000 shift. We attribute the underperformance of OI1000 to label imbalance.
> ```
>
> We hope this addresses your concern.
>
> **W2 (utility of the dataset):** We thank you for raising this question about the utility of our ImageNet++ dataset. While we do consider our SELECT benchmark, rather than our ImageNet++ dataset, to be the core contribution in this paper, we nevertheless believe that there are realistic scenarios where ImageNet++ could prove useful independent of the SELECT benchmark. For instance, authors proposing novel computer vision architectures could use ImageNet++ to ablate the performance of their architectures with respect to pretraining dataset, while controlling for the test dataset.
>
> **W3 (Limitations):** We thank you for this point, and agree that it is important not to overgeneralize from our findings. We will add a limitations section before our conclusion section, emphasizing that we report on only six curation methods, one task (image classification), and one label set (ImageNet). Another limitation of this work is that our modeling of cost is relatively coarse; as more curation methods become available and documentation of curation strategies improves, it will be possible to develop more fine-grained cost estimates.
>
> **W4 (Documentation):** We thank you for the suggestion to provide data documentation in a standard format. We have included a complete datasheet in Appendix Section O, and the NeurIPS checklist. We hope that this addresses your concern.
>
> Thank you again for your excellent comments. We are happy to continue the discussion any time until the end of the discussion period.

---

### Official Review · Reviewer_3uXb · 2024-07-23

**Rating:** 6
**Confidence:** 4
**Correctness:** The claims made in the submission are…
**Clarity:** Please refer to "Review" for my comme…

**Review:**

### Pros:

- **(P1) Significance:** This work addresses a critical and under-explored problem: the systematic evaluation of data curation strategies. This is particularly important as the data curation quality significantly impacts model performance.
- **(P2) Quality:** The provided benchmark (SELECT) covers multiple aspects of model performance, providing a holistic view of data curation quality.  It provides careful analysis of results, training and evaluating over 130 models, offering insights for practitioners and researchers. The released source code, dataset, and full results also promotes the reproducibility.
- **(P3)** **Clarity:** The paper is generally well-written and organized. The authors effectively communicate ideas and methodologies through well-structured sections and informative tables and figures, making it accessible to researchers in computer vision.

### Cons:

- **(C1) The task scope and claims:** The focus is limited to image classification tasks, potentially limiting the generalizability of findings to other computer vision domains, such as detection and segmentation. As such, the title of this paper should also not be “image recognition” but should be specified to “image classification” academic rigor;
- **(C2) Dataset diversity:** The dataset shifts are primarily derived from or related to ImageNet, which may not fully capture the diversity of real-world data distributions.
- **(C3) Metric limitations:** Some proposed metrics like CLIPScore were found to have limited utility in predicting downstream performance. The authors note: "Image and label quality metrics do not provide useful signal for guiding data curation." This highlights the need for further work on developing more informative analytic metrics that correlate better with model performance.

**Strengths:**

**(S1) Addresses a crucial problem:** Data curation is critical but often overlooked in existing computer vision studies. This work brings much-needed attention to evaluating different data curation strategies. The importance of this cannot be overstated, as dataset quality fundamentally impacts model performance, generalization, and fairness. By providing a thorough evaluation framework, this paper enables better decisions about data collection and curation, thus potentially leading to more robust and efficient vision systems.

**(S2) The benchmark quality:**
SELECT covers a wide range of relevant metrics: In-distribution accuracy (ImageNet validation), Out-of-distribution robustness (natural shifts like ImageNet-Sketch, ObjectNet, ImageNet-V2, ImageNet-R, ImageNet-A; synthetic shifts like ImageNet-C and Stylized-ImageNet), Transfer learning (11 diverse tasks inspired by VTAB), and Self-supervised learning (using DINO with varying samples-per-class). This thorough evaluation gives a valuable picture of curation quality, going beyond simple accuracy metrics to assess real-world utility and generalization.

**(S3) The dataset:** The provided IMAGENET++ extends ImageNet with 5 new shifts using different curation strategies: OI1000: OpenImages subset with crowdsourced labels; LA1000(img2img): LAION subset using embedding-based image-to-image search; LA1000(txt2img): LAION subset using embedding-based text-to-image search; SD1000(img2img): Synthetic dataset generated from ImageNet images; SD1000(txt2img): Synthetic dataset generated from ImageNet class names. This creates a valuable resource for studying distribution shifts and curations, enabling experiments on the impact of different data curation strategies.

**(S4) Applications:**  The authors release code, dataset, and full results, enabling verification and extension of this work, which will accelerate progress in this important area of computer vision research.

**Additional Feedback:**

I hope my review helps to further strengthen this work and helps the authors, fellow reviewers, and Area Chairs understand the basis for my recommendation. I look forward to the rebuttal feedback and would be glad to raise my rating if thoughtful responses and improvements are presented.

**Documentation:**

There is sufficient detail on data collection and organization, availability and maintenance, and ethical and responsible use.

**Ethics:**

There are no or only very minor ethical concerns with the submission that warrant further discussion or review.

**Limitations:**

The authors have adequately addressed the limitations and potential negative societal impact of their work.

**Opportunities For Improvement:**

**(L1) Limited task scope and the claims:** The benchmark focuses solely on image classification. Thus, the title of this paper should not be “image recognition” but instead “image classification” for academic rigor. While classification is a fundamental task in visual recognition, expanding to other recognition tasks like object detection and segmentation could provide broader insights. Different tasks may have different sensitivities to data curation strategies, and a more diverse task set could reveal these nuances. I suggest the authors to expand the benchmark to cover additional computer vision tasks beyond classification.

**(L2) Dataset diversity:** While IMAGENET++ introduces new shifts, they are all derived from or related to ImageNet. Including more diverse data sources could potentially help to strengthen the generalizability of findings. This would help assess whether the conclusions hold across different data distributions and application domains. I suggest the authors to include more diverse data sources and domains to further evaluate the generalizability of their findings.

**(L3) Metric limitations:** Some proposed metrics like CLIPScore were found to have limited utility in predicting downstream performance. The authors note: "Image and label quality metrics do not provide useful signal for guiding data curation." This highlights the need for further work on developing more informative analytic metrics that correlate better with model performance. I suggest the authors to develop better metrics that correlate better with downstream performance.

**(L4) The relationship between curation strategies and model architecture/size:** It would be interesting to see if specific data curation methods are more or less effective for different model scales or architectures (e.g., ConvNets vs. ViTs).

**Relation To Prior Work:**

This work has clearly discussed how it differs from previous contributions.

**Summary And Contributions:**

This paper introduces SELECT, a thorough benchmark for evaluating data curation strategies in image classification tasks. The authors provide IMAGENET++, a large-scale dataset that extends ImageNet with 5 additional training data shifts, each employing a different data curation strategy, including Expert curation (baseline ImageNet), Crowdsourced labeling (OpenImages), Synthetic image-to-image generation (Stable Diffusion), Synthetic text-to-image generation (Stable Diffusion), Embedding-based image-to-image search (LAION), and Embedding-based text-to-image search (LAION). This provides a rich testbed for studying data curation methods. The authors evaluate these shifts with a multi-faceted approach, considering: In-distribution accuracy (ImageNet validation set), Out-of-distribution robustness (various natural and synthetic shifts), Transfer learning performance (VTAB-inspired benchmark), and Self-supervised learning capabilities (using DINO).

---

> ### Author Rebuttal · Authors · 2024-08-15
>
> Thank you for your detailed review. We appreciate that you recognize the importance of the problem we address in this paper, and the utility of our benchmark and dataset.
>
> We address your individual questions below:
>
> **L1:** We appreciate your note about the intended scope of our work. We agree that image classification is a fundamental task, and it has historically been predictive of performance on downstream tasks [12]. However, we also agree that it is more accurate to replace “image recognition” with “image classification” in our title, and we promise to do so in the camera-ready version.
>
> **L2:** We appreciate your suggestion about utilizing more diverse pretraining data sources, and agree that this would be a useful future direction. We note, however, that ImageNet is the dominant pretraining dataset, with nearly all timm [13] models using it, making it of particular interest to practitioners. There is also evidence that ImageNet is a strong choice for pretraining, particularly for its scale, and that varying the pretraining data content dramatically (I.e., using synthetic or domain-specific data) while controlling for dataset scale tends to degrade performance on large label sets, which are the focus of SELECT. [1] [2] [3] [4]
>
> **L3:** We appreciate your observation, and agree that superior image and label quality metrics are needed to improve data curation. Following your suggestion, we have added the Inception Score [7] and CMMD [8] to our image quality metrics; please see our global response, section C.  Unfortunately, multimodal data quality research is still in its infancy; considerably more effort has been focused in recent years on boosting the data quality of language-only pretraining datasets. [5] [6] The fact that those efforts have lead to gains in downstream performance suggests that the task is certainly possible; this, too, is an important direction for future research.
>
> **L4:** We thank the reviewer for raising this point. We agree that cross-architecture comparisons would be interesting, as previous work has shown that architecture can have a significant impact on robust accuracy and SSL performance [9] [10]. One challenge with ViTs in particular, and the reason we did not use them in our work, is that they generally underperform ConvNets with access to ImageNet-scale pretraining datasets and smaller. [9] [11]
>
> Thank you again for your excellent comments. We respectfully ask that if you now feel more positively about our paper, to consider increasing your score. We are happy to continue the discussion any time until the end of the discussion period. Thank you!
>
> REFERENCES
>
> [1] https://arxiv.org/pdf/2311.04016
>
> [2] https://arxiv.org/abs/2307.12532
>
> [3] https://arxiv.org/abs/2208.05516
>
> [4] https://arxiv.org/abs/1805.08974
>
> [5] https://github.com/togethercomputer/RedPajama-Data
>
> [6] https://arxiv.org/abs/2407.21783
>
> [7] https://arxiv.org/abs/1606.03498
>
> [8] https://arxiv.org/html/2401.09603v1
>
> [9] https://arxiv.org/abs/2308.03821
>
> [10] https://arxiv.org/abs/2203.03605
>
> [11] https://arxiv.org/abs/2106.10270
>
> [12] https://arxiv.org/abs/2106.08254
>
> [13] https://github.com/huggingface/pytorch-image-models

---

### Official Review · Reviewer_U7mH · 2024-07-24
**Solid Benchmark with Valuable Findings**

**Rating:** 7
**Confidence:** 4
**Correctness:** Claims in this paper are correct, to …
**Clarity:** Paper is well written.

**Review:**

This work provides a robust framework and releases 5 variations of the ImageNet dataset in a convenient manner for evaluating data curation strategies. Findings in terms of effects of curation methods, data scale, and usefulness of commonly-used quality metrics is valuable for practioners and researchers. I recommend this paper for acceptance. The paper could be improved if the authors spent a bit more (maybe in appendix if insufficient pages) to discuss prior related works, as data curation strategies have had fairly rich works recently that should be mentioned.

**Strengths:**

Strengths

1. Wide range of metrics and tasks are evaluated.
2. Extensive range of data mixtures considered.
3. Very useful conclusions drawn in terms of effects of methods, data scale, and usefulness of commonly-used quality metrics.
4. Code is available and documentation is sufficient.

**Additional Feedback:**

N/A

**Documentation:**

Documentation looks solid.

**Ethics:**

No significant ethics concerns noted. But please check ImageNet licensing. I think they have a non-commercial clause that needs to be applied to derivative works.

**Limitations:**

Limitations are well discussed.

**Opportunities For Improvement:**

1. Sec 2.1 Related Work is a bit sparse.
2. Check if ImageNet non-commercial license affects the current work's license.

**Relation To Prior Work:**

Well-discussed relations, but more works could be discussed to provide better context.

**Summary And Contributions:**

This paper investigates different individual data curation strategies for images. To investigate this, the authors curated 5 different data mixtures of ImageNet.  Findings from this work are valuable for developing better strategies for image curation.

---

> ### Author Rebuttal · Authors · 2024-08-15
>
> Thank you for your detailed review. We are happy that you found our conclusions useful and our experiments extensive. We address each of your questions below:
>
> **W1: Related Work**. We thank you for this note about our related works. Owing to space limitations, our original NeurIPS submission divided the related works section in two; an extended related works section can currently be found in Appendix Section I. We will be happy to integrate these two sections in our camera-ready version, where more space is available.
>
> **W2: Licensing**. Thanks for this note. We agree that the commercial licensing terms of our dataset is an important consideration, and we discuss it in Appendix Section B. All the original data we have generated, we release under permissive licenses (CC0 for synth images, MIT for our code), and to the best of our knowledge, there is no current US legal precedent restricting us from freely licensing data generated by diffusion models. We will clarify these distinctions for our final dataset release.
>
> Thank you again for your excellent comments. We are happy to continue the discussion any time until the end of the discussion period.

---

> > ### Comment · Reviewer_U7mH · 2024-08-28
> >
> > I already recommended this paper for an accept -- so nothing new to add.

---

### Author Rebuttal · Authors · 2024-08-15

We thank all of the reviewers for their valuable feedback. Our work introduces SELECT, the first large-scale benchmark of curation strategies for image classification. We appreciate that the reviewers find our experiments comprehensive (U7mH, kQRC), our insights helpful (U7mH, 3uXb, 5d11), and our presentation clear (3uXb). Following the reviewers’ suggestions, we highlight further experimental results in the attached PDF:

* We add per-dataset robustness results for all curation methods (A)
* We provide an updated version of Table 2 with corrected confidence intervals (B)
* We provide more label / image quality metrics (C)

For the camera-ready draft, we will update our paper to include all of these new results.

One meta-observation we made about several comments is that many of them request that we substantially extend our empirical results; I.e., over more tasks, over more architectures, over more metrics, over more pretraining datasets, and over more label sets. While we agree that many of these ablations constitute important directions for future research, as our primary contribution in this work is a benchmark, we want to make it as easy as possible for other researchers to adopt SELECT. Therefore, we think it is important to limit as much as possible its computational and logistical complexity.

We would be very happy to keep the discussion going, addressing any points that remain unclear, or any new suggestions. Thanks again for your suggestions!

(A): Per-Dataset Robustness for all Curation Methods

We provide complete per-dataset robustness results for all of our curation methods on ImageNet-V2, ImageNet-Sketch, ImageNet-Rendition, ImageNet-Adversarial, Objectnet, ImageNet-C, and Stylistic ImageNet.

For the most part, our results are consistent with prior works on distributional robustness, such as [1]; trends in robust accuracy tend to reflect those of base accuracy. One notable outlier is the OI1000 dataset, whose ImageNet-R and ImageNet-A accuracy is higher than would be expected given its base accuracy, most likely due to differences in curation strategies between OpenImages and ImageNet.

(B): Updated Table 2 with Corrected Confidence Intervals

We present an updated version of our Table 2 with corrected confidence intervals.

(C) Additional image and label quality metrics

We add to our existing list the Inception Score [2] and CMMD score [3]. We also provide IN1000-val accuracy for reference. We find that Inception Score is not a reliable predictor of quality as measured by IN1000-Val accuracy, as it favors synthetic SD1000 (txt2img) images over real OI1000 and IN1000 images, which contradicts the commonsense conception of image quality as a measure of realism. CMMD score shows more promise than any other method we have considered, and has the potential to be useful; however, its low score for the OI1000 split is incongruous with other, more reliable measures of label and image quality.

REFERENCES

[1] https://arxiv.org/abs/2007.00644
[2] https://arxiv.org/abs/1606.03498
[3] https://arxiv.org/abs/2401.09603

---

### Comment · Area_Chair_oZXh · 2024-08-28

Dear Reviewers,

Thank you for taking the time to review this submission. :)

This is a gentle reminder regarding the reviewer-author discussion.

Please respond to the author's rebuttal at your earliest convenience, especially if you have any points of disagreement.

The deadline is August 31 at 11:59 PM AoE!

Early discussion is always appreciated.

Best, AC

---

### Decision · Program_Chairs · 2024-09-26

**Decision:**

Accept (Poster)

**Comment:**

We appreciate the authors' efforts in answering reviewers' questions during the rebuttal phase and to a great extent addressed the reviewers' concerns. The paper received diverse ratings: 7,6,7,6. The reviewers are generally positive about the proposed idea SELECT, a thorough benchmark (with a set of metrics and assumptions) for evaluating data curation strategies in image classification tasks. The decision takes into account the paper, the reviews, the rebuttal, and the post-rebuttal reviewer discussion. The AC panel did not find any reason to overturn the majority rating. Based on the above consideration, the AC suggests to accept this paper but the revised version needs to properly include the responses to Reviewers' comments, where possible, the workable links to the dataset and the code.